# High levels of TFAM repress mammalian mitochondrial DNA transcription in vivo

Nina A Bonekamp[1] , Min Jiang[1,2] , Elisa Motori[1,3] , Rodolfo Garcia Villegas[4] , Camilla Koolmeister[4], Ilian Atanassov[5], Andrea Mesaros[6], Chan Bae Park[7], Nils-Göran Larsson[1,4]

**Mitochondrial transcription factor A (TFAM) is compacting mitochondrial DNA (dmtDNA) into nucleoids and directly controls mtDNA copy number. Here, we show that the TFAM-to-mtDNA ratio is critical for maintaining normal mtDNA expression in different mouse tissues. Moderately increased TFAM protein levels increase mtDNA copy number but a normal TFAM-to-mtDNA ratio is maintained resulting in unaltered mtDNA expression and normal whole animal metabolism. Mice ubiquitously expressing very high TFAM levels develop pathology leading to deficient oxidative phosphorylation (OXPHOS) and early postnatal lethality. The TFAM-to-mtDNA ratio varies widely between tissues in these mice and is very high in skeletal muscle leading to strong repression of mtDNA expression and OXPHOS deficiency. In the heart, increased mtDNA copy number results in a near normal TFAM-to-mtDNA ratio and maintained OXPHOS capacity. In liver, induction of LONP1 protease and mitochondrial RNA polymerase expression counteracts the silencing effect of high TFAM levels. TFAM thus acts as a general repressor of mtDNA expression and this effect can be counterbalanced by tissue-specific expression of regulatory factors.**

## Introduction

Treatment of mitochondrial diseases caused by dysfunctional oxidative phosphorylation (OXPHOS) remains a challenge. The clinical variability is substantial as almost any organ and cell type may be affected with varying degrees of severity; with disease onset varying from the neonatal period to late in adult life. Moreover, mutations in two different genomes, that is, nuclear DNA and mitochondrial DNA (mtDNA), can cause OXPHOS defects and thus mitochondrial disease (Rahman, 2020; Russell et al, 2020). The

majority of mitochondrial proteins (~99%), including all that regulate mtDNA maintenance and expression, are encoded in the nucleus and imported into mitochondria. In contrast, mtDNA only contributes 13 proteins that all have essential roles for the function of four of the OXPHOS complexes. Mammalian mtDNA is a compact circular double-stranded genome of about 16.6 kb in size, where each strand undergoes polycistronic transcription followed by RNA processing and maturation to yield 2 ribosomal RNAs (rRNAs), 22 tRNAs, and 11 mRNAs (translated to 13 proteins) (Anderson et al, 1981; Bibb et al, 1981; Gustafsson et al, 2016). In the last 30 yr, substantial progress has been made in understanding the genetic basis of mitochondrial diseases. More than 300 different pathogenic point mutations, deletions and duplications of mtDNA have been identified since the first disease-causing mtDNA mutations were reported (Holt et al, 1988; Wallace et al, 1988). Because of the high copy number of mtDNA, pathogenic mutations may affect all (homoplasmy) or only a subset (heteroplasmy) of all mtDNA molecules in a cell (Sciacco et al, 1994; Taylor & Turnbull, 2005). A biochemical phenotype is induced once a certain threshold level of mutant mtDNA is exceeded. To add to the complexity, mutations in about 300 nuclear genes have been shown to cause mitochondrial disorders by affecting different aspects of mitochondrial function, such as biogenesis of the OXPHOS system, nucleotide transport/synthesis, membrane dynamics, mtDNA maintenance, and mtDNA expression at different levels, including transcription, RNA maturation, and translation (Vafai & Mootha, 2012; Thompson et al, 2020).

The use of animal models has facilitated our understanding of mitochondrial diseases and given insights into a variety of potential treatment options (Russell et al, 2020), for example, important pathophysiological features of mitochondrial myopathy were recapitulated in mice with a disruption in mtDNA expression in skeletal muscle (Wredenberg et al, 2002). Importantly, a substantial increase in mitochondrial mass was demonstrated to partly

[1]Department of Mitochondrial Biology, Max Planck Institute for Biology of Ageing, Cologne, Germany    [2]Zhejiang Provincial Laboratory of Life Sciences and Biomedicine, Key Laboratory of Growth Regulation and Transformation Research of Zhejiang Province, School of Life Sciences, Westlake University, Hangzhou, China    [3]Cologne Excellence Cluster on Cellular Stress Responses in Aging-Associated Diseases (CECAD), Cologne, Germany    [4]Department of Medical Biochemistry and Biophysics, Karolinska Institutet, Stockholm, Sweden    [5]Proteomics Core Facility, Max Planck Institute for Biology of Ageing, Cologne, Germany    [6]Phenotyping Core Facility, Max Planck Institute for Biology of Ageing, Cologne, Germany    [7]Ajou University, Suwon, Republic of Korea

Correspondence: nina.bonekamp@medma.uni-heidelberg.de; nils-goran.larsson@ki.se
Nina A Bonekamp's present address is Department of Neuroanatomy, Mannheim Center for Translational Neuroscience (MCTN), Medical Faculty Mannheim, Heidelberg University, Mannheim, Germany

compensate for reduced OXPHOS function by maintaining the overall ATP production at near-normal levels despite poor function of individual mitochondria in skeletal muscle (Wredenberg et al, 2002). Furthermore, boosting mitochondrial biogenesis has become a promising approach exploited for the treatment of mitochondrial disease (Whitaker et al, 2016). Overexpression of PPAR-γ coactivator 1-α (PGC1α), an important regulator of mitochondrial biogenesis (Puigserver et al, 1998; Wu et al, 1999; Scarpulla, 2002), can improve mitochondrial disease manifestations in mice (Viscomi et al, 2011; Dillon et al, 2012). Because of the complex regulation of PGC1α activity (Fernandez-Marcos & Auwerx, 2011), direct targeting of downstream effectors of the PGC1α signalling cascade, such as the mitochondrial transcription factor A (TFAM) (Virbasius & Scarpulla, 1994), might provide a more directed treatment approach.

TFAM is an essential regulator of mitochondrial function in mammals because of its dual role as a core component of the mitochondrial transcription initiation machinery and as the key factor packaging mtDNA into mitochondrial nucleoids (Falkenberg et al, 2002; Alam et al, 2003; Kukat et al, 2011, 2015; Shi et al, 2012). The TFAM protein consists of two high mobility group-box domains that are separated by a linker and followed by a short C-terminal tail essential for transcription activation (Fisher & Clayton, 1988; Parisi & Clayton, 1991; Dairaghi et al, 1995; Kanki et al, 2004). Each of the two high mobility group box domains causes mtDNA to bend 90° leading to a 180° U-turn when one molecule of TFAM binds mtDNA (Ngo et al, 2011; Rubio-Cosials et al, 2011). Specific binding of TFAM to the two mtDNA promoters results in transcription activation by recruitment of the mitochondrial RNA polymerase (POLRMT) and mitochondrial transcription factor B2 (TFB2M) (Hillen et al, 2017). Nonsequence-specific TFAM binding enables packaging of the mitochondrial genome into protein–DNA complexes to form the mitochondrial nucleoid (Shen & Bogenhagen, 2001; Alam et al, 2003; Kukat et al, 2011, 2015; Bonekamp & Larsson, 2018). Importantly, TFAM levels are known to directly control mtDNA copy number (Larsson et al, 1998; Matsushima et al, 2003; Ekstrand et al, 2004; Kanki et al, 2004) and multiple in vivo studies have shown that TFAM overexpression is beneficial in mouse models with various types of pathology, for example, myocardial infarction, amyotrophic lateral sclerosis, transient forebrain ischemia and age-dependent memory loss (Ikeuchi et al, 2005; Hayashi et al, 2008; Hokari et al, 2010; Morimoto et al, 2012). Increasing the absolute levels of mtDNA has also been shown to improve the function in certain organs in mouse models of mitochondrial disease and premature ageing (Nishiyama et al, 2010; Jiang et al, 2017; Filograna et al, 2019, 2021). Remarkably, the rescue effect has been shown to occur although the proportion of mutant mtDNA stays the same (Jiang et al, 2017; Filograna et al, 2019), thus showing that the absolute levels of wild-type mtDNA play an important role in determining pathogenic effects caused by heteroplasmic mtDNA mutations.

Based on the findings in mouse models, it has been suggested that manipulation of TFAM levels and thus mtDNA copy number is a target for disease intervention. However, increased mtDNA copy number has also been reported to result in enlarged nucleoids, inhibition of mitochondrial transcription and respiratory chain dysfunction (Ylikallio et al, 2010). There are a number of limitations in the abovementioned studies that do not allow a definite conclusion on whether up-regulation of mtDNA copy number by TFAM

overexpression is beneficial or detrimental in vivo. Several studies have relied on a transgenic mouse model expressing the human TFAM cDNA from the synthetic CAG (modified chicken β-actin promoter with CMV-IE enhancer) promoter that causes robust expression of the human TFAM protein in the heart, skeletal muscle, and brain, but barely detectable expression in lung, liver and kidney (Ikeuchi et al, 2005; Hayashi et al, 2008; Hokari et al, 2010; Ylikallio et al, 2010; Morimoto et al, 2012). This tissue-specific expression pattern is likely explained by a positional effect caused by the random genomic integration of the human TFAM cDNA construct. Another complication of this experimental system is that it is heterologous as the human TFAM protein, which only poorly activates mouse mtDNA transcription (Ekstrand et al, 2004), is expressed in the mouse. As a consequence, the mutant mice display an increase in mtDNA levels without increasing OXPHOS or mitochondrial mass (Ekstrand et al, 2004). In contrast, TFAM can stimulate mtDNA expression in an autologous system as it has been reported that import of human TFAM into human mitochondria stimulates mtDNA transcription (Garstka et al, 2003; Maniura-Weber et al, 2004).

Here, we describe a series of mouse models with a moderate or strong overexpression of the endogenous mouse TFAM protein and investigate the effects on mtDNA copy number, mitochondrial gene expression and whole animal physiology. We show that moderate increase in TFAM is well tolerated in the mouse, whereas strong overexpression has deleterious consequences in certain tissues. The most severe phenotype was observed in tissues where the mtDNA copy number remained low, which resulted in a high TFAM to mtDNA ratio that completely abolished mitochondrial gene expression. Thus, although TFAM is essential for transcription initiation, it also acts a general gene repressor of mtDNA expression in mammalian mitochondria in vivo. Modulation of TFAM levels thus serves as a global mechanism to regulate mitochondrial gene expression likely by influencing nucleoid compaction.

## Results

### Moderately increased TFAM levels sustain normal mtDNA expression

To study the consequences of moderately increased TFAM levels in vivo, we generated bacterial artificial chromosome (BAC) transgenic mice harbouring an introduced 203 kb mouse genomic fragment containing Tfam expressed from its endogenous promoter under the control of adjacent regulatory elements (Fig 1A). Three lines of BAC-TFAM transgenic mice derived from independent founders were maintained as heterozygotes on an inbred C57Bl/6N background. BAC constructs are typically randomly inserted into the mouse genome and analysis of the three different founder lines, BAC TG 137 (Figs 1 and S1), BAC TG 188 (Fig S1), and BAC TG 91 (Fig S1) allowed us to rule out effects caused by insertional mutagenesis. Western blot analyses of total tissue extracts revealed a moderate overexpression of TFAM of 1.63-fold in the heart, 1.50-fold in the liver, and 1.49-fold in the skeletal muscle in the BAC TG 137 line compared to control (Fig 1B). The other founder lines also displayed an increase in TFAM protein levels in all tissues

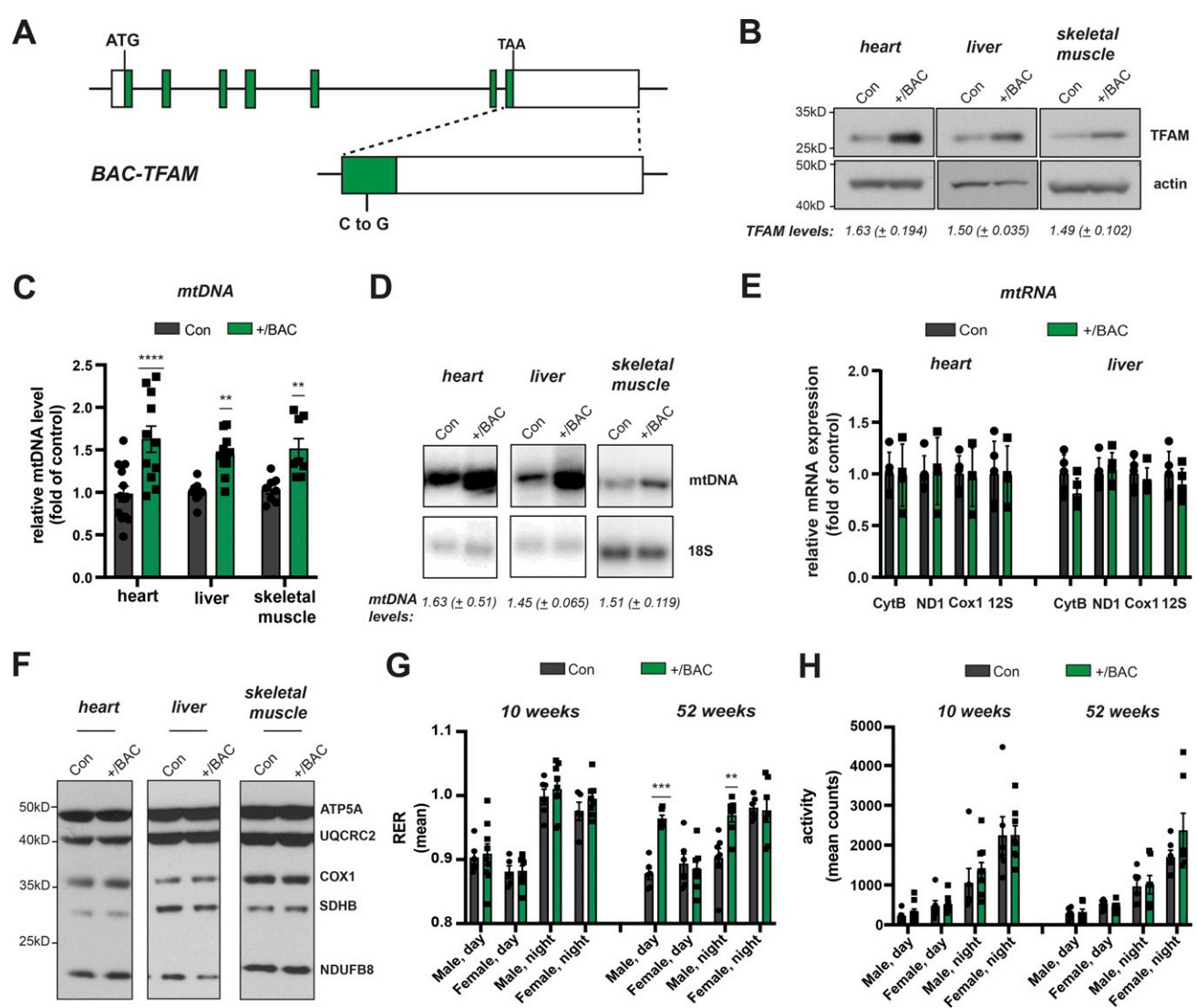

**Figure 1. Moderate increase in TFAM levels leads to increased mtDNA copy number without effects on mitochondrial gene expression.**
**(A)** Overview of the *bacterial artificial chromosome* (BAC) construct expressing mouse Tfam under its endogenous promoter. Green boxes indicate Tfam exons. A neutral point mutation generating a PvuI restriction site was introduced to distinguish the BAC from the Tfam wild-type locus (indicated as C to G in the magnified exon). **(B)** Western blot analysis of TFAM protein levels in the heart, liver, and skeletal muscle whole cell lysates of wild-type (Con) and BAC-TFAM TG 137 (+/BAC) animals. The BAC TG 137 founder line is used in all subsequent experiments and referred to as +/BAC. Actin was used as a loading control. A representative image is shown (n = 3 independent experiments). **(C)** Quantification of steady-state mtDNA levels in the heart, liver, and skeletal muscle of wild-type (Con) and BAC-TFAM (+/BAC) animals. In the case of heart and liver, mtDNA levels were quantified by qPCR using specific probes against COX1 and 18S. Quantification of skeletal muscle mtDNA levels was performed by densitometric analysis of Southern blots. Data are expressed as means ± SEM (n = 9–12 biological replicates for heart and liver; n = 8 for skeletal muscle; $P < 0.01$: **; $P < 0.0001$: ****, two-way ANOVA with Sidak's test for multiple comparisons). **(D)** Southern blot analysis of PstI-digested mtDNA derived from the heart, liver, and skeletal muscle of wild-type (Con) and BAC-TFAM (+/BAC) animals. mtDNA was quantified by radiolabeling with a specific probe against COX1, nuclear DNA was probed with 18S. A representative image is shown (n = 3 independent experiments). **(E)** Analysis of steady-state mitochondrial transcript levels in heart and liver of wild-type (Con) and BAC-TFAM (+/BAC) animals by qRT-PCR. Mitochondrial mRNAs and rRNAs were quantified using specific mouse probes, $\beta$-2-microglobulin was used as a reference gene. (n = 4–5 biological replicates). **(F)** Western blot analysis of steady-state levels of respiratory chain subunits in the heart, liver, and skeletal muscle mitochondrial extracts of wild-type (Con) and BAC-TFAM (+/BAC) animals. A representative image is shown (n = 3 independent experiments). **(G, H)** Phenotyping/energy homeostasis of BAC-TFAM mice aged 10 and 52 wk. Cohorts of BAC-TFAM mice (+/BAC) and wild-type litter mates (Con) were analysed by indirect calorimetry at the age of 10 and 52 wk. Data on respiratory exchange rate (RER, [G]) and activity (sum of ambulatory and fine movement, [H]) are shown (means ± SEM, n = 5–8 biological replicates; $P < 0.01$: **; $P < 0.001$: ***, Two-way ANOVA with Sidak's test for multiple comparisons).
Source data are available for this figure.

investigated, albeit at different levels. In BAC TG 188, a similar level of TFAM expression was observed in heart (1.78-fold), with lower levels in liver and skeletal muscle (Fig S1A and B). In BAC TG 91, we observed a consistent increase in TFAM levels in all tissues, the

highest levels being detected in skeletal muscle (1.68-fold). We next investigated the relative mtDNA copy number in different tissues using quantitative PCR (qPCR) and Southern blotting. In line with the well-established role of TFAM as a key regulator of mtDNA copy

number, increased TFAM levels led to a significant increase in mtDNA up to 1.63-fold of control in the heart, 1.45-fold in the liver, and 1.51-fold in the skeletal muscle in BAC TG 137 mice (Fig 1C and D). The relative mtDNA copy number was similarly increased in the other founder lines and depended on the TFAM levels (Fig S1C and D). Because of the consistent, moderate TFAM increase observed in the different tissues of the heterozygous BAC TG 137 line (Jiang et al, 2017; Filograna et al, 2019), we proceeded to extensively characterize this line, henceforth denoted BAC-TFAM (+/BAC) mice.

Normally, TFAM is present in around 1,000 molecules per mtDNA molecule or 1 TFAM molecule per 16–17 bp of mtDNA in mammalian cells (Kukat et al, 2011). High TFAM-to-mtDNA ratios have been shown to block mitochondrial gene expression in vitro (Farge et al, 2014). In the different tissues of BAC-TFAM mice, the TFAM-to-mtDNA ratio was maintained at the same relative levels as in wild-type mice (1.00 in the heart, 1.03 in the liver, and 0.98 in the skeletal muscle), which should not affect mitochondrial gene expression.

We addressed the steady-state mtDNA transcript levels by quantitative reverse transcription PCR (qRT-PCR) and Northern blotting (Figs 1E and S1E and F). The mitochondrial mRNA, rRNA and tRNA levels were not changed in heart or liver tissue of BAC-TFAM mice (Figs 1E and S1E and F) or in the BAC TG 188 or BAC TG 91 mouse lines (Fig S1E and F), showing that a moderate increase in TFAM expression does not globally affect mtDNA transcription. In line with this, Western blotting of isolated mitochondria from the heart, liver, and skeletal muscle of BAC-TFAM mice showed no change in the expression of OXPHOS subunits (Fig 1F). Furthermore, tandem mass tag (TMT)–based quantitative proteomics on whole tissue lysates of the heart, liver, and spleen of BAC-TFAM mice confirmed that a moderate increase in TFAM protein levels does not affect the total cellular proteome and is therefore unlikely to affect normal tissue function (Fig S2A). Thus, moderate TFAM overexpression causes an increase in mtDNA copy number without affecting overall mtDNA gene expression.

## Moderately increased TFAM and mtDNA levels are well tolerated in vivo

We proceeded to assess physiology in the BAC-TFAM mice at different ages. To this end, we measured energy homeostasis and activity of BAC-TFAM animals and wild-type litter mates at the age of 10 and 52 wk using metabolic cages. This enabled us to investigate their metabolic performance, spontaneous locomotor activities and drinking and feeding behaviour. We did not observe substantial changes in the drinking and feeding behaviour or body weight between wild-type and BAC-TFAM mice of both sexes at 10 and 52 wk of age (Fig S2B–D). The normal weight and food intake in BAC-TFAM mice indicated that they do not suffer from stress, disease conditions or metabolic changes. The measurements of $O_2$ consumption and $CO_2$ production in metabolic cages were used to calculate the respiratory exchange rate (Fig 1G) and the energy expenditure (heat; Fig S2E) that both were within a normal range in BAC-TFAM mice. We observed a significant increase in the respiratory exchange rate in BAC-TFAM males aged 52 wk, which points towards a preferred utilization of carbohydrates in those mice (Fig 1G). This increase did not correlate with a change in food intake or increased activity, as animals did not display any substantial alteration in activity, given as the sum of ambulatory and fine

movements (Fig 1H) and the cumulative distance travelled (Fig S2F). The underlying physiological difference and sex-specificity requires further investigation. Litter sizes were within the normal range of C57Bl6N mice for the BAC-TFAM, BAC TG 188, and BAC TG 91 mouse lines (Fig S2G), showing that the reproductive performance and fertility was not impaired. We thus conclude that BAC-TFAM mice are healthy and indistinguishable from controls and that a moderate increase in TFAM protein levels and mtDNA copy number is very well tolerated in vivo without affecting animal physiology or metabolism.

## Strong TFAM overexpression results in postnatal lethality

Next, we investigated the effects of strong TFAM overexpression on mtDNA copy number and mitochondrial function in vivo. To this end, we generated knock-in mice that can activate the expression of a FLAG-tagged mouse TFAM protein under the control of the CAG promoter in the ROSA26 locus. The ROSA26 locus is known to provide ubiquitous expression and the CAG promoter is a commonly used synthetic promoter driving high levels of transgene expression in almost all cell types of transgenic animals (Niwa et al, 1991; Okabe et al, 1997). Mice heterozygous for the CAG-TFAM allele preceded by a loxP-flanked stop-cassette were mated to mice ubiquitously expressing cre-recombinase (β-actin-cre) to generate mice overexpressing mouse TFAM in all tissues, henceforth denoted CAG-TFAM mice (Fig 2A). The litter sizes of CAG-TFAM mice were normal and the mutant mice were born at the expected Mendelian ratios (Fig S3A). To our surprise, we observed increased lethality of CAG-TFAM mice after postnatal day 16 and all mice died or had to be euthanized before postnatal day 35 (Fig 2B). CAG-TFAM mice were generally smaller and weaker than wild-type litter mates (Fig 2C) and the heart, liver, and kidney were smaller in comparison with wild-type animals (Fig S3B and C). Strong TFAM overexpression thus had a profound effect on animal well-being and we therefore investigated the underlying cause of this detrimental effect on a tissue and cellular level.

Combined cytochrome c oxidase/succinate dehydrogenase enzyme activity (COX/SDH) staining in heart and skeletal muscle of CAG-TFAM mice revealed a pronounced loss of COX-reactive skeletal muscle fibres and a smaller number of COX-negative cardiomyocytes in the heart (Fig 2D). This finding points to a severe mitochondrial dysfunction in muscle tissue. To address a direct link between high TFAM protein levels and the resulting mitochondrial dysfunction, we first ruled out an indirect effect caused by a possible oversaturation of the mitochondrial protein import system. Cytosolic accumulation of mitochondrial precursors has been shown to be hazardous to cellular fitness and can trigger an adaptive response to counteract this cytosolic protein stress (Boos et al, 2019). We reasoned that saturation of the mitochondrial import system, if present, would lead to an increase in mitochondrial precursor proteins in the cytosol and a global decrease in matrix proteins. We detected a similar pattern of aconitase (ACO2) distribution between the cytosol and mitochondria after subcellular fractionation of heart and liver tissue in control and CAG-TFAM animals (Fig S3D). Furthermore, Western blot analysis of mitochondrial proteins with a well-defined and prominent cleavable mitochondrial targeting sequence showed no increase in the levels of non-imported precursors in addition to the mature proteins in the heart, liver, and skeletal muscle tissue extracts, even after

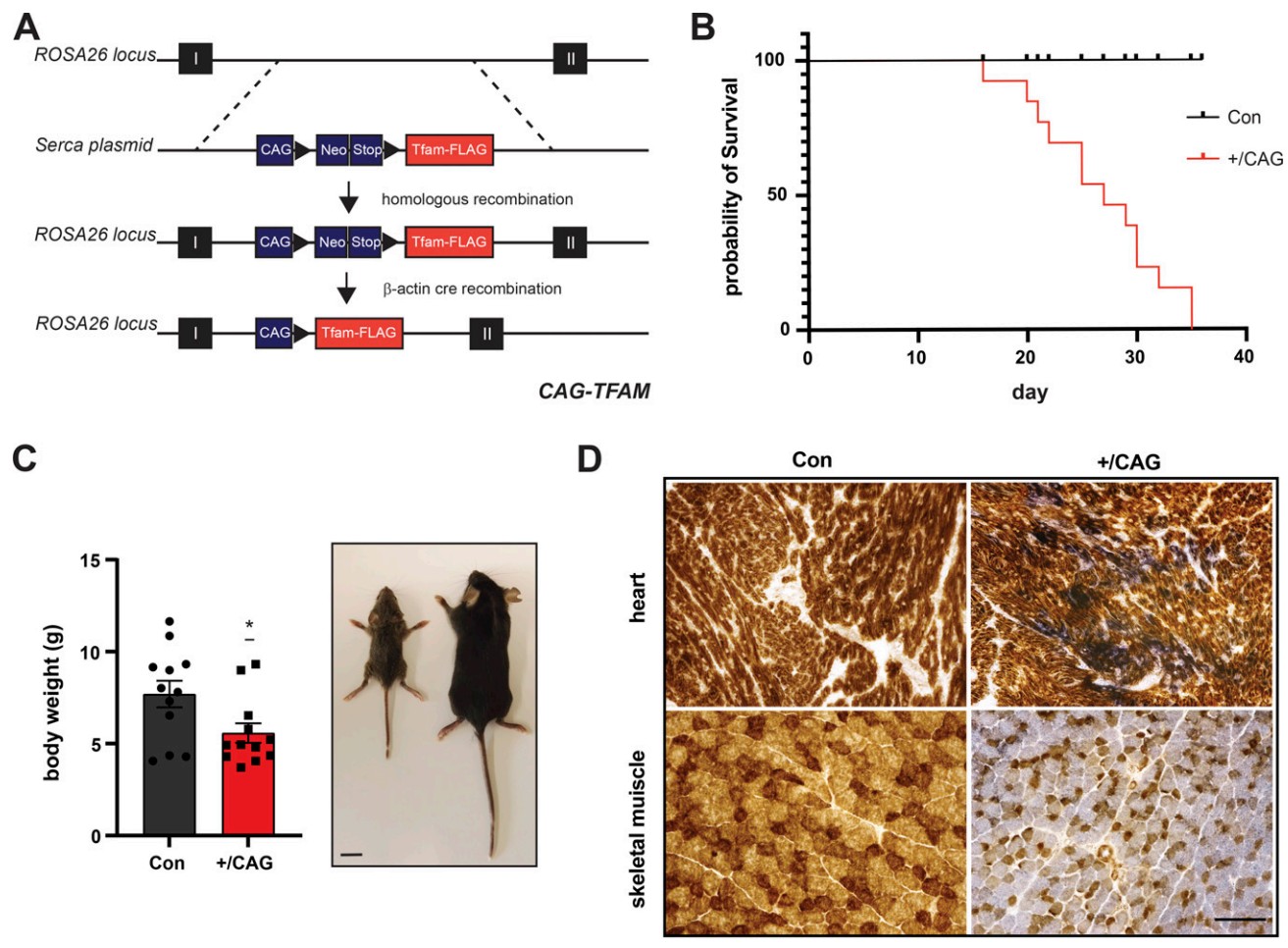

**Figure 2. High TFAM overexpression leads to early postnatal mortality.**
**(A)** Strategy to generate CAG-TFAM mice. A cDNA construct encoding a FLAG-tagged TFAM protein under the control of the CAG promoter was introduced into the ROSA26 locus by homologous recombination. CAG-TFAM mice were generated by crossing to $\beta$-actin cre animals. **(B)** Survival curve of CAG-TFAM mice. Litters yielding CAG-TFAM mice (+/CAG) were observed for 40 d for development and survival compared with control litter mates (Con) (n = 12). **(C)** Body weight of CAG-TFAM mice (+/CAG) compared with control litter mates (Con) at the age of 3 wk. Means ± SEM, n = 12 biological replicates; $P < 0.05$:*, unpaired $t$ test. Scale bar, 1 cm. **(D)** COX/SDH staining of heart (upper panel) and skeletal muscle (lower panel) of CAG-TFAM mice (+/CAG) compared with control litter mates (Con) at the age of 3 wk. Representative images are shown (n = 3 biological replicates). Scale bar, 100 $\mu$m.
Source data are available for this figure.

prolonged exposure of blots (Fig S3E). In addition, the prominent precursor sequences of ATP5A and NDUFA9 were not detectable by quantitative TMT proteomics of whole tissue extracts from the heart, liver, and skeletal muscle (Fig S3F). Our quantitative proteomics data also showed no general defects in mitochondrial protein levels, but rather mildly increased levels of proteins involved in the tricarboxylic acid (TCA) cycle, lipid, and acetyl CoA metabolism, iron-sulphur cluster synthesis, and heme synthesis (Fig S3G). We thus conclude that strong TFAM expression exceeding the normal physiological range can directly shut down mitochondrial function without affecting mitochondrial protein import.

### High TFAM-to-mtDNA ratios abolish mtDNA expression in skeletal muscle

We proceeded to analyse the underlying molecular basis for the observed mitochondrial dysfunction in different tissues from CAG-

TFAM mice. Mitochondrial metabolism and function is different between tissues, and mitochondria from different organs therefore differ in their biosynthetic capacity and ultrastructural appearance (Vafai & Mootha, 2012). Western blotting of total tissue extracts from CAG-TFAM mice revealed that the relative TFAM expression was increased 4.46-fold in heart and 3.84-fold in skeletal muscle in comparison with wild-type mice (Fig 3A and B). These levels much exceeded the TFAM levels observed in BAC-TFAM mice (Fig 1B). Although the relative TFAM overexpression levels were comparable in both heart and skeletal muscle in CAG-TFAM mice, the mitochondrial dysfunction was much more severe in skeletal muscle than in heart (Fig 2D). Changes in the TFAM-to-mtDNA ratio affect mitochondrial function in vitro, so we proceeded to analyse potential differences in mtDNA levels between the two tissues by qPCR and Southern blotting. We observed markedly increased mtDNA levels in heart (6.35-fold of control), whereas the mtDNA copy number (0.96-fold of control) was not changed in skeletal muscle

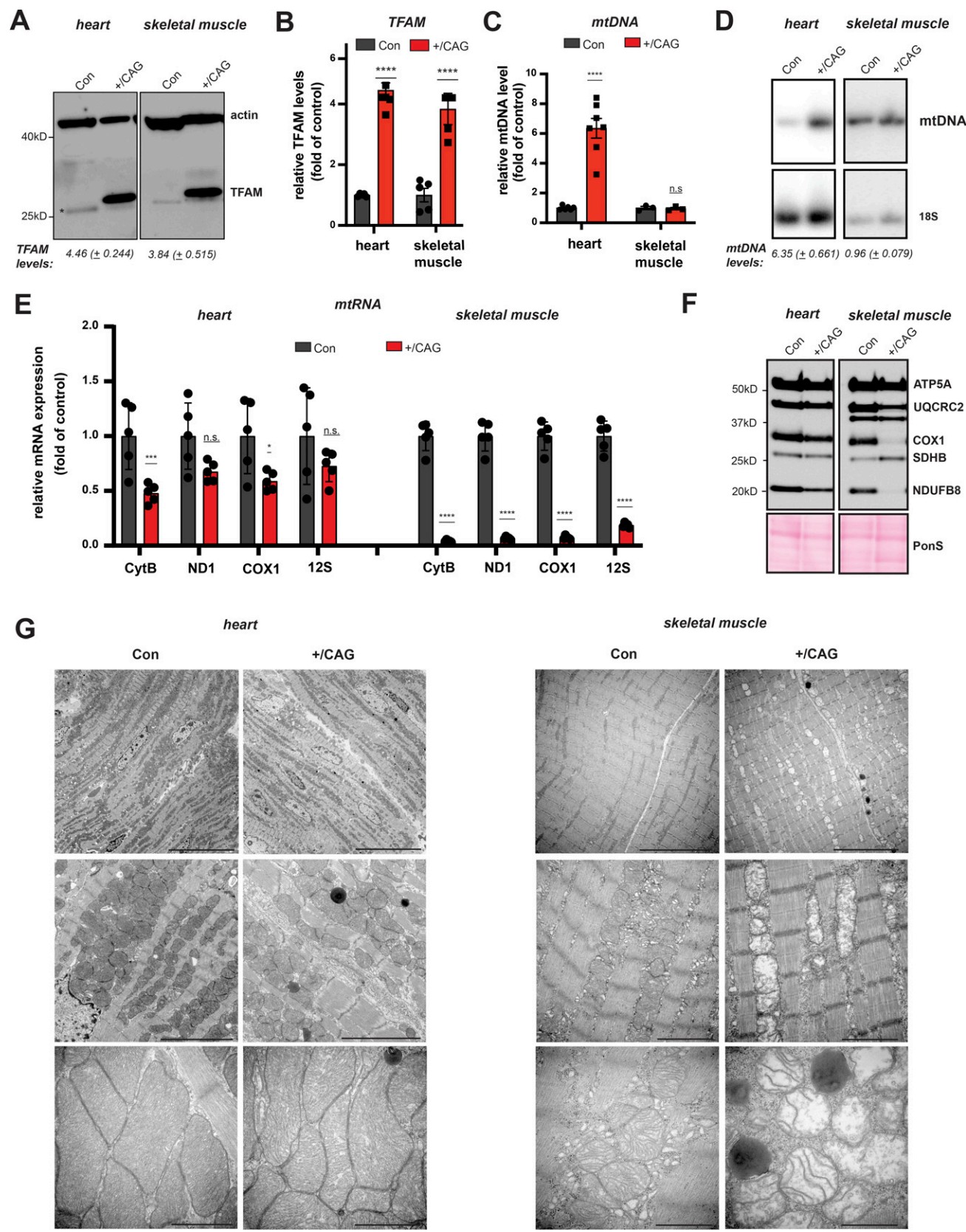

despite the strong increase in TFAM protein levels (Fig 3C and D). The relative TFAM-to-mtDNA ratio was therefore markedly increased to 3.96-fold (3.84 TFAM:0.96 mtDNA) in skeletal muscle. These findings indicate that nucleoids in skeletal muscle of CAG-TFAM mice may be too saturated with TFAM and too compacted to allow sufficient transcription for normal mtDNA expression. qRT-PCR of mitochondrial steady-state transcript levels indeed confirmed a severe depletion of mitochondrial mRNAs and rRNAs in skeletal muscle of CAG-TFAM mice (Fig 3E). We also observed a moderate decrease in CytB and Cox1 transcript levels in the heart, despite the strong mtDNA copy number increase (Fig 3D). In line with this, OXPHOS protein levels were only slightly decreased in heart mitochondria, whereas Western blotting of OXPHOS protein subunits in isolated skeletal muscle mitochondria revealed a strong depletion of the mitochondrially encoded COX1 subunit of Complex IV and the nucleus-encoded NDUFB8 subunit of Complex I. The levels of the UQCRC2 subunit of Complex III were mildly decreased (Fig 3F).

To determine the effects of altered TFAM-to-mtDNA ratios on mitochondrial and tissue organization, we performed transmission electron microscopy of tissue sections from heart and skeletal muscle (Fig 3G). In heart, we observed an altered mitochondrial morphology with less densely packed cristae, however, the overall organization and structure of heart tissue remained unchanged (Fig 3G, left). Cristae arrangement and nucleoid distribution were recently shown to be intrinsically linked by a highly ordered co-organization (Stephan et al, 2019). Thus, we investigated mitochondrial nucleoids in heart sections by stimulated emission depletion (STED) microscopy in CAG-TFAM mice. The nucleoid numbers in CAG-TFAM mice were markedly increased in line with the observed increase in mtDNA copy number, but the nucleoids formed extensive clusters (Fig S4A and B). This increased nucleoid clustering was absent in cardiomyocytes of BAC-TFAM mice (Fig S4A and B). Importantly, STED analysis demonstrated that the diameters were very similar to nucleoid diameters in control and BAC-TFAM mice (Fig S4C and D), consistent with our previous findings showing that a nucleoid typically contains a single copy of mtDNA (Kukat et al, 2011, 2015). The increased nucleoid clustering might thus account for less dense packing of mitochondrial cristae observed in heart tissue. In contrast to the heart, mitochondrial ultrastructure was drastically changed in skeletal muscle tissue of CAG-TFAM mice, with a strong decrease in the number of cristae and a disordered appearance (Fig 3G, right). These changes also affected the ultrastructure of the normally highly ordered and densely packed skeletal muscle fibres. Our results thus show that a balanced TFAM-to-mtDNA ratio is critical to sustain mtDNA expression and OXPHOS for normal tissue function.

## Tissue-specific responses to high TFAM levels

To gain more insight into the tissue-specific responses to high TFAM levels, we proceeded to perform quantitative TMT-based proteomics of different tissues of CAG-TFAM mice. In line with our previous observations, we found a very severe depletion in the levels of OXPHOS subunits of complexes I, III, and IV and mito-ribosomal subunits in skeletal muscle, whereas the effects were milder in heart (Fig 4A and B). Interestingly, whole tissue quantitative proteomics revealed very different mitochondrial responses to marked TFAM overexpression (Fig S5A). A significant fraction of all mitochondrial proteins was down-regulated in the severely affected skeletal muscle, whereas mitochondrial protein levels were maintained in the less affected heart of CAG-TFAM mice (Fig S5A). In contrast, we observed increased levels of a large fraction of all mitochondrial proteins, concomitant with stable or even mildly increased levels of OXPHOS proteins in liver of CAG-TFAM mice on a whole cell level (Figs 4A and S5A). The TFAM protein levels in liver were lower than the levels in heart or skeletal muscle (Fig 4C). Interestingly, the LONP1 protease was among the most up-regulated proteins in liver of CAG-TFAM mice, but it was not increased in heart or skeletal muscle (Fig 4D). Other studies have reported that LONP1 degrades excess TFAM that is not bound to mtDNA (Matsushima et al, 2010; Lu et al, 2013). This free TFAM has been reported to be degraded by LONP1 after being marked by site-specific phosphorylation (Lu et al, 2013). However, despite the very high TFAM levels in skeletal muscle of CAG-TFAM mice (Fig 3A and B) we did not observe increased TFAM phosphorylation by Phostag gel electrophoresis experiments (Fig S5B). A similar observation was made in liver (Fig S5C). We cannot rule out that more sensitive techniques may be required to detect post-translational modifications regulating only a subset of nucleoids.

Thus, in contrast to heart and skeletal muscle, liver appears to respond to increased TFAM levels by a compensatory response including the up-regulation of LONP1 and components of the mitochondrial transcription machinery such as the mitochondrial RNA polymerase (POLRMT) (Fig 4C and D).

---

**Figure 3. High TFAM-to-mtDNA ratios abolish mtDNA expression in skeletal muscle.**
**(A)** Western blot analysis of TFAM protein levels in heart and skeletal muscle whole cell lysates of CAG-TFAM (+/CAG) mice. Litter mates were used as controls (Con). Actin was used as a loading control. The asterisk indicates the lower wild-type TFAM band in control mice as opposed to the overexpression of the FLAG-tagged version of TFAM in CAG-TFAM mice. A representative image is shown (n = 2 independent experiments). **(B)** TFAM protein levels in heart and skeletal muscle whole cell lysates of CAG-TFAM animals (+/CAG) and control litter mates were quantified by densitometry and are expressed as folds of control (means ± SEM, n = 4–5 biological replicates; $P < 0.0001$: ****, two-way ANOVA with Sidak's test for multiple comparisons). **(C)** Quantification of steady-state mtDNA levels in heart and skeletal muscle of CAG-TFAM (+/CAG) animals and control litter mates (Con). mtDNA levels were quantified by qPCR using specific probes against COX1 and 18S. Data are expressed as means ± SEM (n = 6–7 biological replicates for heart; n = 3 for skeletal muscle; n.s., non-significant, $P < 0.0001$: ****, two-way ANOVA with Sidak's test for multiple comparisons). **(D)** Southern blot analysis of PstI-digested mtDNA derived from heart and skeletal muscle of CAG-TFAM (+/CAG) animals and control litter mates (Con). mtDNA was quantified by radiolabeling with a specific probe against COX1, nuclear DNA was probed with 18S. A representative image is shown (n = 3 independent experiments). **(E)** Analysis of steady-state mitochondrial transcript levels in heart and skeletal muscle of CAG-TFAM (+/CAG) animals and control litter mates (Con) by qRT-PCR. Mitochondrial mRNAs and tRNAs were quantified using specific mouse probes, $\beta$-2-microglobulin was used as a reference gene. (n = 5 biological replicates, $P < 0.05$: *; $P < 0.001$:***; $P < 0.0001$: ****, two-way ANOVA with Sidak's test for multiple comparisons). **(F)** Western blot analysis of steady-state levels of respiratory chain subunits in heart and skeletal muscle mitochondrial extracts of CAG-TFAM (+/CAG) animals and control litter mates (Con). A representative image is shown (n = 3 independent experiments). **(G)** Representative images of fixed heart (left) and skeletal muscle (right) tissue from control (con) and CAG-TFAM animals at 16 d of age analysed by transmission electron microscopy. For each genotype, six biological replicates were analysed. Scale bars, heart: 20 (upper panels), 5 (middle), 1 $\mu m$ (lower panels); skeletal muscle: 10 (upper panel), 2 (middle), and 1 $\mu m$ (lower panel).
Source data are available for this figure.

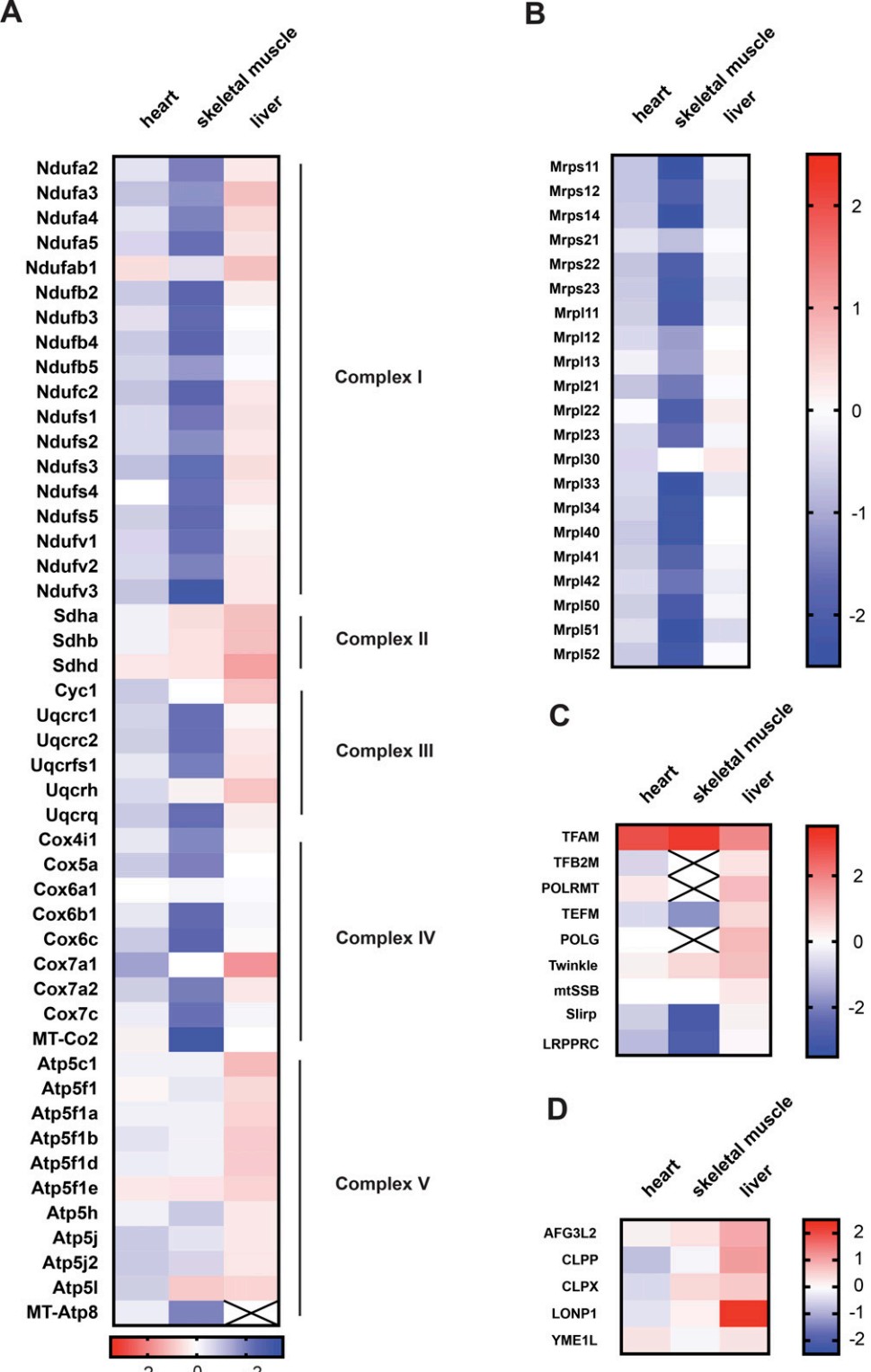

**Figure 4. Tissue-specific responses to high TFAM levels.**

**(A, B, C, D)** Heat map illustrating the log$_2$ fold-change in protein levels of OXPHOS subunits (A), mitoribosomal subunits (B), components of the mtDNA expression machinery (C), and ATP-dependent mitochondrial proteases (D) in the heart, skeletal muscle, and liver of CAG-TFAM mice compared to litter mates. Heat map: minimum (−2), blue; maximum (2), red.

## Mitochondrial gene expression is maintained in liver tissue

Next, we analysed the effects of the observed compensatory responses on mitochondrial gene expression in liver. Western blotting of total tissue extracts from CAG-TFAM mice confirmed the lower increase in TFAM levels in liver (2.10-fold of control, Fig 5A and B). We observed no significant increase in mtDNA levels in liver (Fig 5C and D). The relative TFAM-to-mtDNA ratio is thus ~2.41-fold

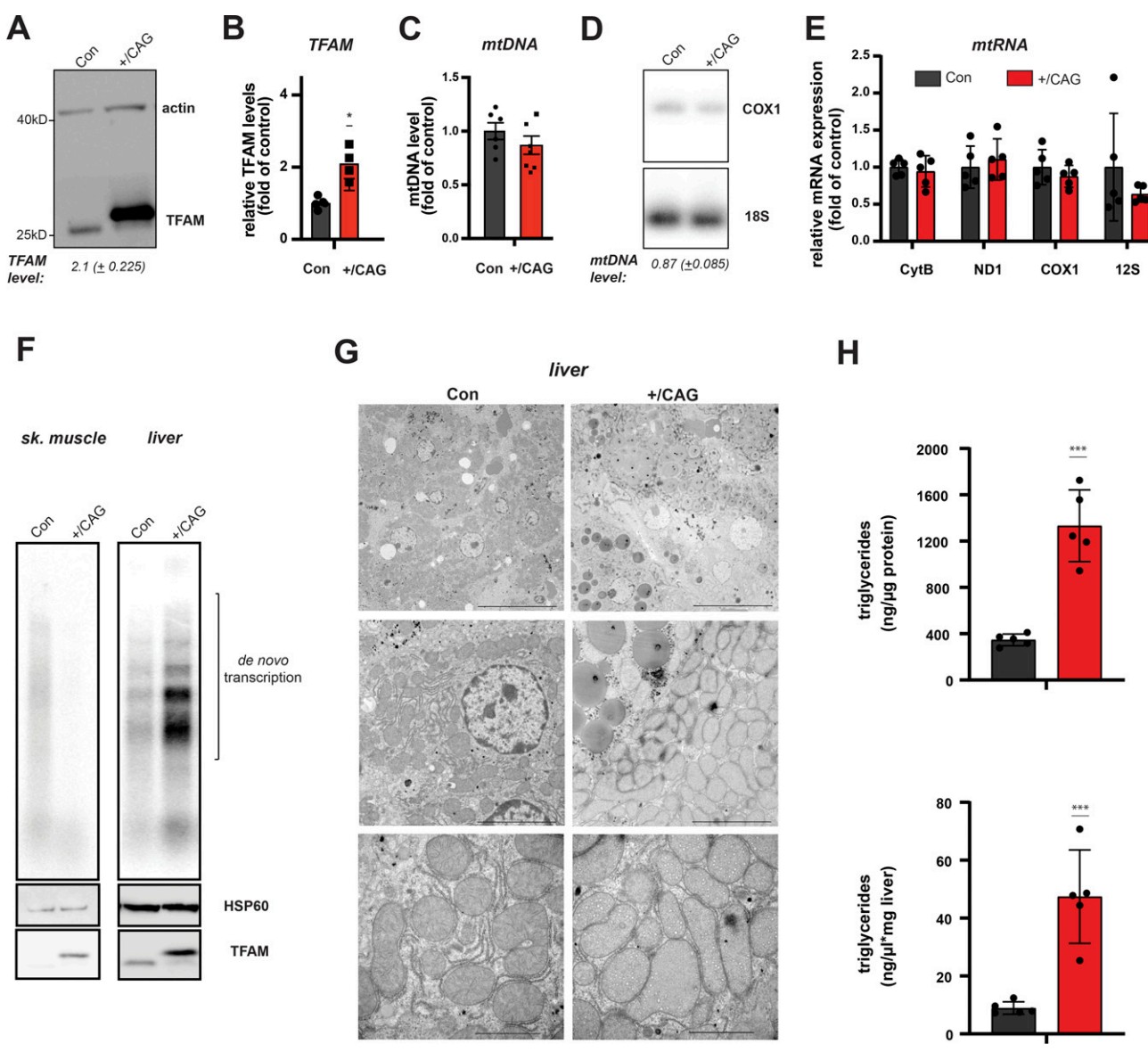

**Figure 5. mtDNA expression is maintained despite high TFAM levels in liver.**
**(A)** Western blot analysis of TFAM protein levels in liver whole cell lysates of CAG-TFAM (+/CAG) mice. Litter mates were used as controls (Con). Actin was used as a loading control. A representative image is shown (n = 2 independent experiments). **(B)** TFAM protein levels in control and CAG-TFAM animals were quantified by densitometry and are expressed as folds of control (means ± SEM, n = 4–5 biological replicates; P < 0.05: *, two-way ANOVA with Sidak's test for multiple comparisons). **(C)** Quantification of steady-state mtDNA levels in liver tissue of CAG-TFAM (+/CAG) animals and control litter mates (Con). mtDNA levels were quantified by qPCR using specific probes against COX1 and 18S. Data are expressed as means ± SEM (n = 6–7 biological replicates). **(D)** Southern blot analysis of PstI-digested mtDNA derived from heart and skeletal muscle of CAG-TFAM (+/CAG) animals and control litter mates (Con). mtDNA was quantified by radiolabeling with a specific probe against COX1, nuclear DNA was probed with 18S. A representative image is shown (n = 3 independent experiments). **(E)** Analysis of steady-state mitochondrial transcript levels in liver tissue of CAG-TFAM (+/CAG) animals and control litter mates (Con) by qRT-PCR. Data are expressed as means ± SEM (n = 5 biological replicates). **(F)** De novo RNA synthesis in skeletal muscle and liver mitochondria isolated from CAG-TFAM (+/CAG) mice and control litter mates. Mitochondria were pulse labelled for 1 h. Mitochondrial HSP60 was used as a loading control. A representative image is shown (n = 2–3 independent experiments). **(G)** Representative images of fixed liver tissue from control (Con) and CAG-TFAM animals at 16 d of age analysed by transmission electron microscopy. For each genotype, six biological replicates were analysed. Scale bars, 20 (upper panels), 5 (middle), 2 μm (lower panels). **(H)** Measurement of triglyceride content. Approximately 100 mg of liver tissue was homogenized and triglycerides were quantified using the triglycerides quantification kit (Sigma-Aldrich). Data are expressed as means ± SEM (n = 5 biological replicates; P < 0.001:***, unpaired t test).
Source data are available for this figure.

higher than in controls in liver of CAG-TFAM mice and this circumstance should lead to increased nucleoid compaction and reduced mtDNA expression in line with previous in vitro findings (Farge et al, 2014; Kukat et al, 2015). However, the steady-state levels

of mitochondrial transcripts were normal in liver of CAG-TFAM animals as assessed by qRT-PCR (Fig 5E). Interestingly, in organello transcription assays of isolated liver mitochondria showed a much higher transcription level compared with wild-type

mitochondria, whereas skeletal muscle mitochondria had very low de novo transcription (Fig 5F), consistent with the observed significant increase in the levels of POLRMT in liver but not in heart in our proteomic dataset (Fig 4C). It is thus apparent that an increased TFAM-to-mtDNA ratio and subsequent repression of mtDNA expression can be compensated for in some tissues, like liver. However, on an ultrastructural level we observed a marked change in mitochondrial morphology with vesicle-like cristae appearance accompanied by an increase in lipid droplets in the liver (Fig 5G). Determination of triglyceride levels relative to liver protein or tissue (Fig 5H) confirmed an accumulation of lipids after strong TFAM overexpression. This points to a dysregulation of lipid metabolism, potentially as a side effect to the strong induction of mitochondrial proteases and other adaptive responses observed in liver.

To summarize, the data we present here show that a moderate increase in TFAM protein levels of 1.5-fold will not interfere with nucleoid morphology, mtDNA replication, mitochondrial gene expression and animal physiology. However, this moderate mtDNA copy number increase is sufficient to ameliorate disease phenotypes caused by heteroplasmic mtDNA mutations (Jiang et al, 2017; Filograna et al, 2019). In contrast, strong ubiquitous TFAM overexpression leads to postnatal lethality and mitochondrial dysfunction. The TFAM-to-mtDNA ratio in combination with tissue-specific regulation is critical factors to determine whether mtDNA expression will be maintained at levels sufficient to sustain OXPHOS and organ function.

## Discussion

TFAM is the major structural protein of the mammalian nucleoid, covering mtDNA with a ratio of 1 TFAM molecule per 16–17 bp of mtDNA (Alam et al, 2003; Kukat et al, 2011). Importantly, TFAM is sufficient to fully compact mtDNA into nucleoids by a series of events starting with single TFAM molecules cooperatively binding to mtDNA in patches (Kaufman et al, 2007; Farge et al, 2014; Kukat et al, 2015). TFAM compacts mtDNA by cross-strand binding and loop formation, thus condensing the mammalian mtDNA with a contour length of ~5 μm into a nucleoid structure with a diameter of ~100 nm (Brown et al, 2011; Kukat et al, 2011). For comparison, an even higher degree of DNA compaction is observed in the nucleus, especially in the condensed states of mammalian chromosomes. Wrapping of DNA around the histone octamer globally conceals nuclear promoters and makes them inaccessible to the preinitiation complex, rendering the core nucleosome a general nuclear gene repressor (Lorch et al, 1987; Han & Grunstein, 1988). Nuclear transcription is thus shut off by default and a complex machinery acts to specifically activate gene expression, for example, by recruitment of the transcription initiation machinery, binding of transcriptional activators, histone remodelling, and addition of posttranslational histone modifications (Kornberg & Lorch, 2020).

In contrast to nuclear transcription, mtDNA transcription may be constitutively active. Quantitative assessment of mtDNA transcription in wild-type mouse embryonic fibroblasts has shown that the vast majority of nucleoids are transcribed (Ramos et al, 2019). Transcripts from mtDNA form mitochondrial RNA granules next to

almost all nucleoids as determined by FISH analyses of individual transcripts or BrU labelling of nascent transcripts (Ramos et al, 2019). It is thus important to recognize that regulation of mitochondrial gene expression needs to ensure promoter-specific initiation of mtDNA transcription. Besides its role in mtDNA compaction (Kaufman et al, 2007; Kukat et al, 2011, 2015), TFAM also binds specifically to the mtDNA promoters and is critical for recruiting POLRMT and TFB2M for initiation of transcription (Hillen et al, 2017) (Fig 6). Although POLRMT on its own is poor at melting double-stranded DNA (dsDNA), it can initiate transcription when binding to single-stranded (ssDNA) in vitro (Wanrooij et al, 2008). Transient strand-separation of mtDNA may result from mtDNA replication or supercoiling tension, but typically, ssDNA is covered by the mitochondrial single-stranded binding protein (mtSSB) and therefore not accessible to binding by POLRMT and initiation of unspecific transcription (Fusté et al, 2010) (Fig 6). In line with this, there is a drastic increase in unspecific transcription initiation of mtDNA in vivo in a conditional mouse knockout model lacking mtSSB in heart (Jiang et al, 2021). The promoter-specific initiation of mtDNA transcription is thus dependent on both TFAM and mtSSB.

Experiments with in vitro reconstituted mitochondrial nucleoids have shown that the TFAM-to-mtDNA ratio determines whether the nucleoid is in an open (active) or a compacted (inactive) state (Kaufman et al, 2007; Farge et al, 2014; Kukat et al, 2015). Variations in local TFAM protein levels may therefore shift the equilibrium between open and inactive nucleoids and thereby control mtDNA gene expression at the level of compaction. We show here that TFAM can also serve as a general repressor of mtDNA gene expression in vivo. We observed a substantial decrease in mitochondrial gene expression in the skeletal muscle of CAG-TFAM mice where the TFAM-to-mtDNA ratio vastly exceeds the ratio found in normal skeletal muscle. Because of the increased TFAM protein levels, the equilibrium likely shifts and may force the nucleoids to adapt a hyper-packaged state which abolishes mitochondrial gene expression (Fig 6). In heart sections of CAG-TFAM mice, we observed an increase in the number of nucleoids consistent with the increased mtDNA copy number. The nucleoids of CAG-TFAM mice were of similar size as nucleoids of control or BAC-TFAM mice, although they formed extensive clusters. Because of technical challenges, we could not directly address nucleoid size by STED microscopy of tissue sections from skeletal muscle and liver. Still, oversaturation of the system with TFAM is likely the basis for the observed skeletal muscle phenotype in CAG-TFAM mice. In agreement with this model, forced expression of Abf2p, the TFAM homologue in budding yeast, leads to eventual loss of mtDNA due to exclusion of replication factors by excessive nucleoid compaction (Zelenaya-Troitskaya et al, 1998). However, increased mtDNA copy number under respiring growth conditions resulted in a more balanced Abf2p-to-mtDNA ratio and more open, transcriptionally active nucleoids in the budding yeast (Kucej et al, 2008). Here, we observe markedly increased mtDNA levels in heart tissue of CAG-TFAM mice after high levels of TFAM expression. This leads to a far more balanced TFAM-to-mtDNA ratio in heart, which allows for continued mtDNA expression. The postnatal development of cardiomyocytes involves substantial mtDNA replication during the first 4 wk of postnatal life (Ramos et al, 2019) which may explain the observed increase in mtDNA levels in heart. However, we observe

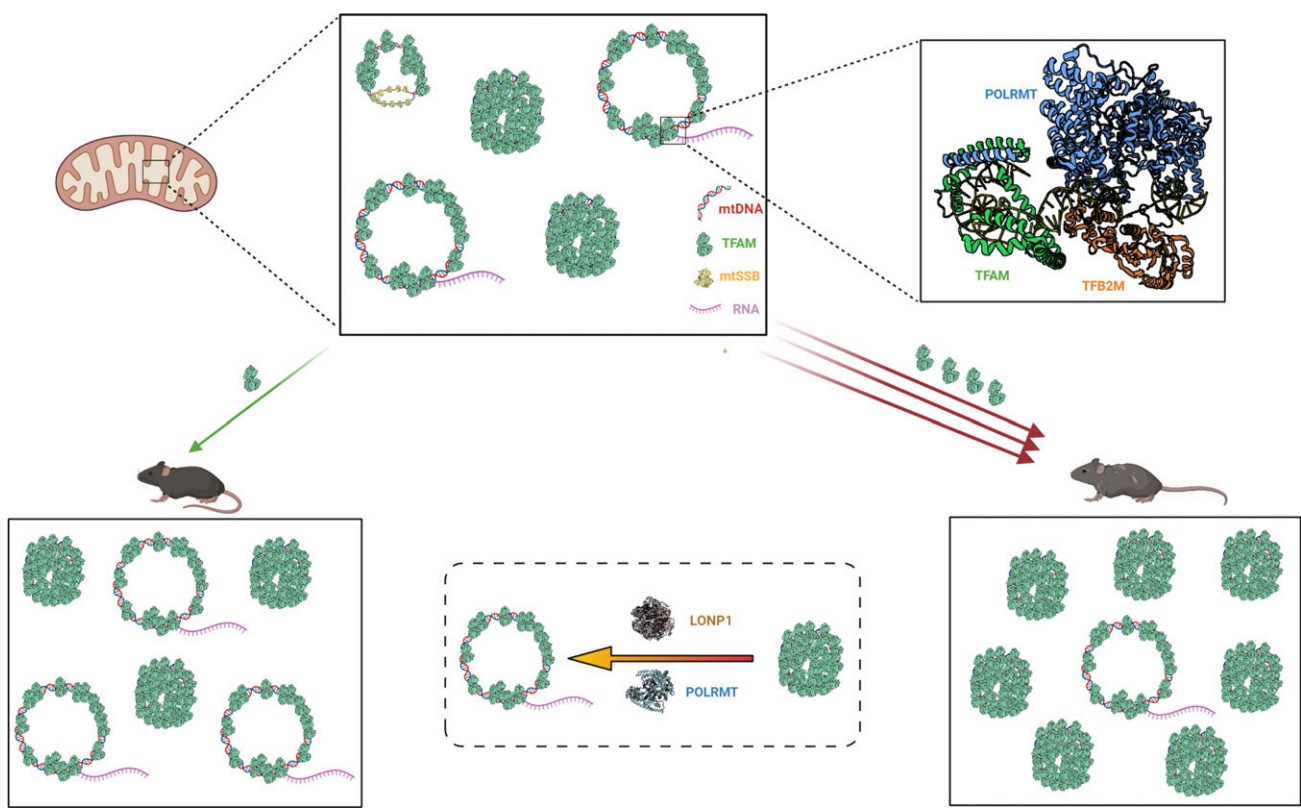

**Figure 6. Regulation of mtDNA expression by TFAM levels in vivo.**
An overview of the proposed regulation of mtDNA expression by TFAM-induced compaction of mitochondrial nucleoids. Open and compacted mitochondrial nucleoids are schematically depicted as circles. The other icons are explained in the figure. This figure was generated using Biorender.com (PDB entries: 3SPA, 7KSM, 2DUD, 6ERP, 3TMM) (Ngo et al, 2011; Ringel et al, 2011; Hillen et al, 2017).

milder adverse effects of TFAM overexpression in the heart and we cannot rule out that more detrimental effects in heart may occur with increasing age despite initial compensation mechanisms (Ghazal et al, 2021). TFAM-to-mtDNA ratios also vary between the different stages of Xenopus oocyte development (Shen & Bogenhagen, 2001), thus TFAM-mediated repression may represent an important, conserved mechanism that controls mtDNA expression in response to the metabolic needs during development and for appropriate function of differentiated tissues.

Our data indicate that regulatory mechanisms may counteract TFAM-induced mitochondrial gene repression in certain tissues. We observed a marked increase in LONP1 levels in the liver of CAG-TFAM mice. In contrast to skeletal muscle, TFAM protein levels were lower and the TFAM-to-mtDNA ratio was more balanced in liver, which may explain the near-normal mitochondrial transcript levels. The TFAM-to-mtDNA ratio observed in liver should be sufficient to repress mtDNA transcription according to in vitro findings (Farge et al, 2014), but the action of LONP1 in vivo may lead to local changes in TFAM levels that may result in decreased compaction of the nucleoid at the promoter regions, which may allow recruitment of POLRMT and TFB2M for initiation of transcription. We observed an induction of both LONP1 and POLRMT expression in liver, which may keep a larger fraction of nucleoids in an open, active state to allow promoter-specific transcription initiation. The tissue-specific induction of the LONP1 protease in liver, but not in skeletal muscle or

heart, points to the existence of cell-type-specific responses regulating TFAM levels. However, this adaptation may come with a cost as we observed accumulation of triglycerides in the liver, which points to dysregulation of lipid metabolism, possibly as a consequence of the up-regulation of mitochondrial proteases interfering with metabolic pathways.

Our data highlight the importance of maintaining TFAM-to-mtDNA ratios within a certain physiological interval to allow proper regulation of mtDNA transcription. Local changes in TFAM levels may provide an important mechanism that controls the switch between repression and activation of mtDNA gene expression by regulation of promoter-specific transcription initiation. Possible treatment strategies to counteract mitochondrial dysfunction by induction mitochondrial biogenesis, for example, by activation of PGC1α, or selective TFAM overexpression must take into account the resulting TFAM-to-mtDNA ratios. Based on the findings presented here, we conclude that modulation of TFAM levels to increase mtDNA copy number by 1.5-fold provides a safe intervention, whereas strong overexpression of TFAM may result in excessive repression of mitochondrial gene expression, nucleoid clustering and ultrastructural changes of mitochondria in different tissues. We thus conclude that the TFAM-to-mtDNA ratio provides an essential mechanism that control mtDNA gene expression in vivo and that this ratio must be maintained or proper function oxidative phosphorylation.

# Materials and Methods

## Animal models and housing

We generated BAC-TFAM mice as previously described (Jiang et al, 2017). Founder mice carrying *Tfam* BAC DNA were identified by PvuI restriction analysis of mouse genomic DNA. Germ line transmission was established from different founder mice and the offspring was kept as heterozygous stocks by breeding to C57Bl/6N mice. Three different mouse lines (BAC TG 188, TG 137 and TG 91) corresponding to three different founders were kept for molecular analysis.

The plasmid construction was performed as previous described (Sterky et al, 2011). In brief, a mouse Tfam CDS sequence without a stop codon was cloned and a flag sequence was added at its C terminus. This sequence was cloned into the Serca plasmid backbone containing CAG promoter-loxP-Neo-Westfal Stop cassette-loxP sites (kind gift from T Wunderlich). The targeting vector was linearized and electroporated into mouse embryonic stem (ES) cells. Southern blotting was used to identify the positive colonies from transformed ES cell lines, which were used for blastocyst injection, and germ line transmission was obtained by mating chimeric mice to C57BL/6N mice.

All mice used in this study had an inbred C57Bl/6N background and were housed in standard individually ventilated cages in a 12 h light/dark cycle in controlled environmental conditions (21°C ± 2°C, 50% + 10% relative humidity). Mice were fed a normal chow diet (ssniff) and water ad libitum. The study was approved by the Landesamt für Natur, Umwelt und Verbraucherschutz, Nordrhein-Westfalen, Germany (reference numbers 84-02.04.2015.A103, 84-02.50.15.004 and 84-02.04.2016.A420) and performed in accordance with the recommendations and guidelines of the Federation of European Laboratory Animal Science Associations.

## Analysis of energy homeostasis

Energy homeostasis and activity of BAC-TFAM animals and age-matched wild-type mice was measured at the age of 10 and 52 wk using metabolic cages (Phenomaster; TSE systems). Before the actual experiments, animals were acclimatized to the different housing conditions in training cages for 3–4 d. Mice were housed in metabolic cages for 3–4 d with data being collected for a time span of 48 h. Differences in $O_2$ and $CO_2$ levels were measured and were used to calculate $O_2$ consumption, $CO_2$ production, respiratory exchange rate, and energy expenditure (heat). Disruption of light beams simultaneously documented animal activity.

## Isolation of total protein and mitochondria from tissues

Animals were euthanized by cervical dislocation, and isolated tissues were cleaned and directly snap-frozen in liquid nitrogen. Total proteins were extracted from ground tissue powder using 2× SDS-sample buffer (100 mM Tris, pH 6.8, 4% SDS, 20% glycerol, and 200 mM DTT) supplemented with complete protease and phosphatase inhibitor cocktail (Roche). The protein concentration was determined using the RCDC assay (Bio-Rad) and BSA as a standard.

Mitochondria were isolated from mouse tissues using differential centrifugation. Briefly, fresh tissues were cut, washed with ice-cold PBS and homogenized in standard mitochondrial isolation buffer containing 320 mM sucrose, 10 mM Tris–HCl, and 1 mM EDTA by using a Potter S pestle (Sartorius). The homogenate was centrifuged at 1,000*g* for 10 min at 4°C. The supernatant was collected and centrifuged at 10,000*g* for 10 min at 4°C. Resulting crude mitochondrial pellets were resuspended in mitochondrial isolation buffer supplemented with complete protease inhibitor cocktail (Roche). Protein concentration was determined using the Bradford method (Sigma-Aldrich) and BSA as a standard.

## Gel electrophoresis and Western blotting

For standard gel electrophoresis, protein samples (30 $\mu$g/lane) were mixed with 2× NuPAGE LDS sample buffer supplemented with 200 mM DTT and resolved using commercially available 10% or 4–12% NuPAGE Bis-Tris gels and MOPS or MES buffer (Invitrogen) including protein standards (Spectra Multicolor Broad Range; Thermo Fisher Scientific). Proteins were transferred on nitrocellulose membranes using wet tank blotting (25 mM Tris, 192 mM glycine, and 20% methanol) at 4°C for 2 h at 400 mA or overnight at 80 mA.

For detection of TFAM phosphorylation, the tissues were lysed in 20 mM Tris–HCl, 150 mM NaCl, 1 mM EDTA, 1 mM EGTA, 1% Triton X-100, and protease inhibitors. 100 $\mu$g of whole tissue lysate were treated with or without $\lambda$ phosphatase (NEB) according to the manufacturer's instructions. Samples were precipitated using TCA and protein pellets were resuspended in 2× SDS sample buffer supplemented with $\beta$-mercaptoethanol. Samples were boiled for 5 min before loading on $Zn^{2+}$ Phostag gels using a neutral-pH Bis-Tris buffering system (15 $\mu$g/lane) (Kinoshita & Kinoshita-Kikuta, 2011). Briefly, 10% polyacrylamide Bis-Tris gels were prepared with the addition of both Phostag acrylamide (50 $\mu$M final) and $ZnCl_2$ (100 $\mu$M final) and run in buffer containing 100 mM MOPS, 100 mM Tris, 5 mM sodium bisulfite, and 0.10% SDS. Phostag gels were soaked three times in transfer buffer containing 5 mM EDTA for 10 min to remove $Zn^{2+}$ before wet transfer overnight.

For standard and Phostag PAGE, membranes were blocked in 5% milk-1× Tris-buffered saline-0.1% Tween 20 (TBST) for at least 1 h at room temperature. Membranes were subsequently incubated with primary antibodies diluted in 5% milk-TBST overnight at 4°C, washed in TBST and incubated with HRP-conjugated secondary antibodies for 2 h at room temperature. After washing with TBST, immunodetection was performed by enhanced chemiluminescence (GE Healthcare) using either photo film or the Fujifilm LAS 400 imaging system (Fujifilm). The following antibodies were used: rabbit polyclonal anti-TFAM (Abcam), mouse monoclonal anti-actin (Abcam), Total OXPHOS Rodent Western Blotting antibody cocktail (Abcam), mouse monoclonal anti-tubulin (Sigma-Aldrich), rabbit anti-Tubulin (Cell Signaling), rabbit anti-vinculin (Abcam), mouse monoclonal anti-UQCRFS1/RISP (Abcam), mouse monoclonal anti-COX5a (Invitrogen/ThermoFisher Scientific), mouse monoclonal anti-VDAC (Millipore), rabbit polyclonal anti-ACOT2 (Proteintech), mouse monoclonal anti-cytochrome C (Abcam), rabbit anti-HSP60 (Cell Signaling), sheep anti-mouse IgG (GE Healthcare), and donkey anti-rabbit (GE Healthcare). For densitometry of

protein levels, intensity of protein bands was analysed using either the MultiGauge Software (Fuji) on LAS imaging files or ImageJ on scanned photo film.

## DNA-isolation and determination of mtDNA levels by qPCR

Total DNA was extracted and purified using the Puregene Core A Kit (QIAGEN) following the manufacturer's instructions including RN-Ase treatment. The purity and quantity of DNA were evaluated with the NanoDrop 2000 (Thermo Fisher Scientific) and 5 ng/$\mu$l DNA were analysed by qPCR. qPCR was carried out using the Taqman 2× Universal PCR mastermix, No Amperase UNG (Applied Biosystems), and commercially available Taqman assay probes for mouse mitochondrial (COX1, Mm04225243_g1) and nuclear DNA (18S, Hs99999901_s1).

## Southern blotting

2 $\mu$g of total DNA were digested with PstI restriction enzyme overnight at 37°C. Digested DNA was precipitated with ethanol, resuspended in water and DNA fragments were separated overnight by 0.6–0.8% agarose gel electrophoresis. Southern blotting to nitrocellulose membranes (Hybond-N+; GE healthcare) was carried out as described before (Kauppila et al, 2018). Hybridisation was carried out using radiolabelled probes against mouse COX1 and 18S to detect mitochondrial and nuclear DNA, respectively.

## RNA-isolation, Northern blotting, cDNA synthesis, and qRT-PCR

Total RNA was isolated from snap-frozen tissue samples using TRIzol Reagent (Thermo Fisher Scientific) following the manufacturer's instructions and subsequently DNAse-treated (TURBO DNA-free kit; Thermo Fisher Scientific). If only intended for Northern blotting, the RNA pellet was directly resuspended in deionized formamide. Alternatively, total RNA was isolated using the Direct-zol RNA Miniprep Kit (ZymoResearch) including the DNAse digestion step. Purity and quantity of RNA were evaluated with the NanoDrop 2000 (Thermo Fisher Scientific). Northern blot analysis was carried out as previously described (Kauppila et al, 2018). Briefly, 2 $\mu$g of RNA were separated on 1.2% paraformaldehyde-agarose gel in MOPS running buffer and transferred to nitrocellulose membranes (Hybond-N+) by capillary transfer. Detection of mitochondrial mRNAs and rRNAs was carried out using $^{32}$P-dCTP-labeled probes, mitochondrial tRNAs were detected using specific oligonucleotides labelled with $\gamma$-$^{32}$P-ATP. 18S was hybridised as a loading control. cDNA was synthesized using the High-Capacity RNA-to-cDNA Kit (Applied Biosystems). qRT-PCR was carried out using the Taqman 2× Universal PCR mastermix, No Amperase UNG (Applied Biosystems) and commercially available Taqman Assay probes for mouse mitochondrial transcripts (CytB, Mm04225271_g1; ND1, Mm04225274_s1; COX1, Mm04225243_g1; 12S, Mm04260177_s1; Life Technologies). Transcript quantities were normalized to $\beta$-2-microglobulin used as a reference gene transcript (Mm00437762_m1; Life Technologies).

## *In organello* transcription

Mitochondria were isolated from fresh heart and liver tissue by differential centrifugation (1 × 1,000$g$, 1 × 10,000$g$). For in organello transcription experiments, freshly purified mitochondria (500 $\mu$g) were washed three times in incubation buffer (25 mM sucrose, 75 mM sorbitol, 10 mM Tris–HCl, 10 mM K$_2$HPO$_4$, 100 mM KCl, 0.05 mM EDTA, 1 mM ADP, 5 mM MgCl$_2$, 10 mM glutamate, 2.5 mM malate, and 1 mg/ml BSA, pH 7.4). Washed mitochondria were resuspended in 500 $\mu$l of incubation buffer and supplemented with 50 $\mu$Ci of $\alpha$-$^{32}$P-UTP (Hartmann Analytic). Samples were incubated at 37°C for 1 h. Afterwards, mitochondria were pelleted, resuspended in incubation buffer containing 0.2 mM UTP and incubated for 10 min at 37°C. Mitochondria were subsequently washed twice in cold wash buffer (10% glycerol, 0.15 mM MgCl$_2$, and 10 mM Tris–HCl, pH 6.8). An aliquot of the mitochondria (10 $\mu$l) was taken as a loading control. Then, RNA was extracted using TRIzol (Ambion) following the manufacturer's recommendations and precipitated overnight at −20°C. The purified RNA was loaded onto a formaldehyde–agarose gel and blotted as a northern blot. The membrane (Hybond-N+; GE Healthcare) was exposed to a phosphorimager screen. Loading controls were run on 10% NuPage gels using MOPS buffer and equal loading was assessed by immunoblotting against HSP60.

## Quantitative proteomics

Sample preparation from 10 mg of grinded tissue powder was performed as described (Busch et al, 2019) with slight modifications: tryptic peptides were eluted from STAGE tips with 40% acetonitrile (ACN) 0.1% formic acid (FA). 4 $\mu$g of desalted peptides were labelled with TMTs (TMT10plex, Cat. no 90110; Thermo Fisher Scientific) using a 1:20 ratio of peptides to TMT reagent. All 10 samples per time point were labelled in one TMT batch. TMT labelling was carried out according to manufacturer's instruction with the following changes: 0.8 mg of TMT reagent was re-suspended with 70 $\mu$l of anhydrous ACN, dried peptides were reconstituted in 9 $\mu$l 0.1 M Tetraethy-lammoniumbromid (TEAB) to which 7 $\mu$l TMT reagent in ACN was added to a final ACN concentration of 43.75%, after 60 min of incubation the reaction was quenched with 2 $\mu$l 5% hydroxylamine. Labelled peptides were pooled, dried, resuspended in 200 $\mu$l 0.1% FA, split into two samples, and desalted using home-made C18 STAGE tips (Rappsilber et al, 2003). One of the two halves was fractionated on a 1 × 150 mm ACQUITY column, packed with 130 Å, 1.7 $\mu$m C18 particles (Cat. no. SKU: 186006935; Waters), using an Ultimate 3000 UHPLC (Thermo Fisher Scientific). Peptides were separated at a flow of 30 $\mu$l/min with a 96 min segmented gradient from 1% to 50% buffer B for 85 min and from 50% to 95% buffer B for 3 min, followed by 8 min of 95% buffer B; buffer A was 5% ACN, 10 mM ammonium bicarbonate (ABC), buffer B was 80% ACN, 10 mM ABC. Fractions were collected every 3 min, and fractions were pooled in two passed (1 + 17, 2 + 18 … etc.) and dried in a vacuum centrifuge (Eppendorf). Dried fractions were resuspended in 0.1% FA separated on a 50 cm, 75 $\mu$m Acclaim PepMap column (Cat. no 164942; Thermo Fisher Scientific) and analysed on a Orbitrap Lumos Tribrid mass spectrometer (Thermo Fisher Scientific) equipped with a field asymmetric ion mobility spectrometry (FAIMS) device (Thermo Fisher Scientific) that was operated in two compensation voltages, −50 and −70 V. Alternatively, peptides were separated on a 25 cm, 75 $\mu$m PicoFrit column (New Objective) packed with 1.9 $\mu$m ReproSil-Pur media (Dr. Maisch) and analysed on an Orbitrap Fusion Tribrid mass spectrometer (Thermo Fisher Scientific). Synchronous

precursor selection based MS3 was used for TMT reporter ion signal measurements. Peptide separations were performed on an EASY-nLC1200 using a 90 min linear gradient from 6 to 31% buffer; buffer A was 0.1% FA, buffer B was 0.1% FA, 80% ACN. The analytical column was operated at 50°C. Raw files were split based on the FAIMS compensation voltage using FreeStyle (v. 1.6; Thermo Fisher Scientific).

Proteomics data were analysed using MaxQuant (Cox & Mann, 2008) (version 1.6.10.43). Raw proteomics data was searched against the mouse proteome database from Uniprot, downloaded in September 2018. The isotope purity correction factors, provided by the manufacturer, were included in the analysis. Differential expression analysis was performed using limma (Ritchie et al, 2015) in R (R Core Team, 2018). The raw data, database search results, and the data analysis workflow and results were deposited to the ProteomeXchange Consortium via the PRIDE partner repository (Perez-Riverol et al, 2019) with the dataset identifier PXD023050.

## Quantification of triglycerides

Approximately 100 mg of liver tissue was homogenized in a 1 ml solution of 5% IGEPAL CA-630 (Sigma-Aldrich). The homogenate was incubated at 90°C for 5 min, then cooled to room temperature. The heating–cooling cycle was repeated to solubilize all triglycerides. Samples were centrifuged at top speed for 2 min, and the supernatant was collected. The samples were diluted 10-fold with water before the quantification, then 5–30 $\mu$l of the samples were used with the triglycerides quantification kit (Sigma-Aldrich) following the manufacturer's instructions.

## COX/SDH staining

COX/SDH double staining was performed as previously described (Matic et al, 2018). Briefly, fresh heart and skeletal muscle tissues were dissected and immediately frozen in isopentane chilled with liquid nitrogen. Tissues were further cryo-sectioned into sections (10 $\mu$m for skeletal muscle and 7 $\mu$m for heart), mounted on slides and left to air dry briefly. Freshly prepared buffer A (0.8 ml of 5 mM 3,3'-diaminobenzidine tetrahydrochloride, 0.2 ml of 500 $\mu$M cytochrome c, and 10 $\mu$l of catalase) was added to the slides. After incubation for 60 min at 37°C, slides were washed three times by 0.1 M phosphate buffered saline, pH 7.0. Then freshly prepared buffer B (0.8 ml 1.875 mM of nitroblue tetrazolium, 0.1 ml 1.3 M of sodium succinate, 0.1 ml 2.0 mM phenazine methosulphate, and 10 $\mu$l of 100 mM sodium azide) was applied and incubated for 30 min at 37°C. Slides were washed three times with 0.1 M phosphate buffered saline, pH 7.0, dehydrated and mounted for bright-field microscopy.

## Transmission electron microscopy

Liver, skeletal muscle and heart tissue were cut into small pieces and fixed in 2.5% glutaraldehyde, 1% paraformaldehyde, and 0.1 M phosphate buffer, pH 7.4 at room temperature for 1 h, followed by 24 h at 4°C. After the fixation, the specimens were rinsed in a buffer containing 0.1 M sodium phosphate and subsequently post-fixed in 2% osmium tetroxide at 4°C for 2 h. The specimens were then stepwise dehydrated in ethanol followed by acetone and finally embedded in LX-112 (Ladd Research Industries). Ultra-thin sections

(60–80 nm) from longitudinal parts were cut and examined in a Hitachi HT7700 (Hitachi High-Tech) at 80 kV. Equipped with a 2kx2k Veleta CCD camera (OSIS). Digital images at a final magnification of 2,500×, 5,000×, 10,000×, 20,000×, and 40,000× were randomly acquired from sections of the tissues.

## gSTED analysis of nucleoids in heart tissue

### Immunohistochemistry
Freshly isolated hearts were fixed in 4% PFA, cryopreserved in 30% sucrose and frozen in optimal cutting temperature compound (OCT). 10-$\mu$m-thick cryosections were cut and air-dried before proceeding with fluorescent immunohistochemistry. Sections were permeabilized in 0.5% Triton X-100/PBS and unspecific binding of antibodies was prevented by incubation in blocking buffer (3% BSA/PBS). The following primary antibodies in blocking buffer were then applied overnight at +4°C: mouse IgM anti-DNA (Progen), rabbit anti-TOM20 (Santa Cruz). After washing, the following secondary antibodies were then applied: goat anti-mouse IgM Alexa 594, and donkey anti-rabbit Alexa 488. Nuclei were counterstained with DAPI.

### Imaging
Imaging was performed with a Leica TCS SP8 gated STED (gSTED) microscope, with a white light laser and a 93× objective lens (HC PL APO CS2 93× GLYC, NA 1.30). For confocal images of mitochondria (TOM20) and DNA, Z-stacks were taken by exciting the fluorophores at 488 and 594 nm, respectively, and Hybrid detectors collected fluorescent signals. STED images of DNA channel were obtained with a 775-nm depletion laser. 2D confocal and gSTED images were acquired sequentially with the optical zoom set to obtain a voxel size of 17 × 17 nm. Excitation was provided at 594 nm and Hybrid detectors collected signal. Gating between 0.3 and 6 ns was applied. Performance of the microscope and optimal depletion laser power were tested as previously described (Nicholls et al, 2018).

## Image processing

Raw images were first deconvolved with the Huygens software. Image panels were created with Photoshop (Adobe); no digital manipulation was applied, except for adjustment of brightness and contrast.

# Data Availability

All relevant data generated and analysed are included in the article. For the proteomic data, the raw data, database search results, and the data analysis workflow and results were deposited to the ProteomeXchange Consortium via the PRIDE partner repository with the dataset identifier PXD023050. Source data are provided with this article. Further requests should be directed to the corresponding authors.

# Supplementary Information

# Acknowledgements

Work described here was supported by the Max Planck Society, the Swedish Research Council (2015-00418 to N-G Larsson), the Swedish Cancer Foundation (to N-G Larsson), the Knut and Alice Wallenberg foundation (to N-G Larsson), the Deutsche Forschungsgemeinschaft (SFB1218/A06 to N-G Larsson), European Research Council (Advanced grant 2016–741366 to N-G Larsson), and the ALF agreement (agreement between the Swedish state and some county councils on cooperation on basic education of doctors, medical research, and the development of health care) to N-G Larsson (SLL2018.0471). M Jiang was supported by the Westlake Education Foundation. E Motori was supported by the SFB1218 Advanced PostDoc Grant. We thank Nadine Hochhard for excellent technical assistance. We thank Avan Taha and Xinping Li from the Proteomic Core Facility and Martin Purrio of the Phenotyping Core Facility for support. We thank the Karolinska Center for Transgene Technologies for technical support in generating CAG-TFAM and the TFAM BAC mouse lines. The serca backbone plasmid was a kind gift from T Wunderlich. We thank the FACS & Imaging Core Facility at the Max Planck Institute for Biology of Ageing and the CECAD Imaging Facility. We thank Lars Haag and the Department of Laboratory Medicine at Karolinska for assistance in transmission electron microscopy. Fig 6 was created using Biorender.com using the protein data bank entries 3SPA, 7KSM, 6DUD, 6ERP, and 2TMM.

## Author Contributions

NA Bonekamp: conceptualization, data curation, formal analysis, supervision, validation, investigation, visualization, methodology, project administration, and writing—original draft, review, and editing.
M Jiang: data curation, investigation, and methodology.
E Motori: data curation, formal analysis, investigation, and methodology.
R Garcia Villegas: data curation, formal analysis, and investigation.
C Koolmeister: methodology.
I Atanassov: data curation, formal analysis, and investigation.
A Mesaros: formal analysis and investigation.
CB Park: methodology.
N-G Larsson: conceptualization, resources, supervision, funding acquisition, project administration, and writing—original draft, review, and editing.

## Conflict of Interest Statement

N-G Larsson is a scientific founder and holds stock in Pretzel Therapeutics, Inc.

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
