## [Reviewer comments · Life Science Alliance]

Life Science Alliance

High levels of TFAM repress mammalian mitochondrial DNA transcription in vivo

Nina Bonekamp, Min Jiang, Elisa Motori, Rodolfo Garcia Villegas, Camilla Koolmeister, Ilian Atanassov, Andrea Mesaros, Chan Bae Park, and Nils-Göran Larsson

DOI: <https://doi.org/10.26508/lsa.202101034>

Corresponding author(s): Nils-Göran Larsson, Karolinska Institutet and Nina Bonekamp, Max Planck Institute for Biology of Ageing; Mannheim Center for Translational Neuroscience

Review Timeline:

Submission Date:	2021-01-25
Editorial Decision:	2021-03-24
Revision Received:	2021-07-29
Editorial Decision:	2021-08-17
Revision Received:	2021-08-20
Accepted:	2021-08-20

Transaction Report:

March 24, 2021

Re: Life Science Alliance manuscript #LSA-2021-01034-T

Prof. Nils-Göran Larsson
Karolinska Institutet
Department of Medical Biochemistry and Biophysics
Solnavägen 9
Stockholm, Stockholm 17165
Sweden

Dear Dr. Larsson,

Thank you for submitting your manuscript entitled "High levels of TFAM repress mammalian mitochondrial DNA transcription in vivo" to Life Science Alliance (LSA). The manuscript was assessed by expert reviewers, whose comments are appended to this letter.

We apologize for this unusual and extended delay in getting back to you. As you will note from the reviewers' comments, the reviewers are interested in these findings, but reviewer 1 and 2 have also raised some questions that should be clarified with text and additional experiments prior to further consideration of the manuscript at LSA. We, thus, encourage you submit a revised version of the manuscript to LSA that addresses all of the reviewers' points.

Thank you for this interesting contribution to Life Science Alliance. We are looking forward to receiving your revised manuscript.

Sincerely,

Shachi Bhatt, Ph.D.

Executive Editor

Life Science Alliance

<https://www.lsjournal.org/>

Interested in an editorial career? EMBO Solutions is hiring a Scientific Editor to join the international Life Science Alliance team. Find out more here -

https://www.embo.org/documents/jobs/Vacancy_Notice_Scientific_editor_LSA.pdf

B. MANUSCRIPT ORGANIZATION AND FORMATTING:

Reviewer #1 (Comments to the Authors (Required)):

The paper of Bonekamp et al deals with a very interesting topic: the effect of the TFAM level on the mitochondrial gene expression in vivo.

The authors, use transgenic mice overexpressing TFAM and analyse the impact of the increased

TFAM level on mitochondrial gene expression in different organs.

They found that a moderate overexpression of TFAM maintains a normal TFAM/mtDNA ratio in all tissue examines (heart, skeletal muscle liver) and has no substantial effect on mitochondrial function and on the whole animal metabolism.

Then the authors overexpress TFAM at a higher level and find a strong repression of mitochondrial function in skeletal muscle, associated with a strong increase of the TFAM/mtDNA. In heart and in liver however the TFAM/mtDNA ratio is almost normal and OXPHOS capacity is maintained. The authors argue that in liver the higher tissue specific expression of LONP1 protease counteracts the TFAM increase, suggesting the existence of tissue specific regulatory factors that counterbalance the TFAM-induced repression of mtDNA transcription.

The data reported in the paper are convincing and clearly reported and in general the paper is well written. However there are a number of points that need to be considered before the paper can be considered for the publication

MAJOR POINTS

1. A first basic point concerns the way in which the data are presented. Initially the authors, describe the BAC-TFAM mice and show the level of TFAM, mtDNA, mtRNA respiratory complexes and then report data on the metabolism of the whole animal. Then they describe the CAG-TFAM mice. In this case the data are presented in a very different order; in particular data concerning mtDNA, mtRNA and respiratory complexes are shown almost at the end of the paper. I think that this may confuse the reader and therefore I believe that for more clarity the authors should present the data in a more ordered way.

In addition, and most importantly, the authors have carried out a complex experimental plan reporting a variety of the effects of TFAM level in three tissues (heart, skeletal muscle and liver), While I certainly appreciate this effort, that has produced a considerable amount of data, I found that in some cases the data are rather fragmentary; some results are presented only in one or two tissues and this makes difficult to assess properly the significance of the paper. Therefore the authors should complete the experiments in the points specified below (see point 5 and point 6)

2. The authors report the steady state level of mitochondrial transcripts (Fig 1E,F, EV1E,F and Fig 3D,E) by using the Northern blot method. I think that the authors should present (at least in some cases) also data coming from qRT-PCR analyses. In particular the data shown in Fig 1E (liver) and 3D (skeletal muscle) are of low quality: bands are very faint and it is difficult to make any quantitative assessment.

3. In Fig 1I the authors report the respiratory exchange rate (RER) in BAC-TFAM mice and conclude that they were similar to control rats. This is true for all samples tested except for the male /day and male/night at 52 weeks, in which the value of BAC-TFAM mice is higher than controls. The authors should explain this difference.

4. To rule out that the mitochondrial dysfunction in BAC-TFAM mice was caused by an overaccumulation of mitochondrial precursors, the authors perform a western blot of mitochondrial matrix protein of liver and skeletal muscle and conclude that there is no accumulation of non-imported precursors (Fig EV3D). The authors state that the bands shown in the figure refer to unprocessed precursors; I think that to justify this statement they should show some MW marker and the corresponding mature products. Moreover, in Fig EV3 the authors present data on the distribution of ACOT2 in the heart demonstrating a similar pattern of distribution in mitochondria and cytosol of control and CAG-TFAM animals. I wonder why the authors have performed different tests in the three tissues examined; in my opinion (unless there is a specific reason) it would have been preferable to test the same protein in the three organs.

5. In Fig EV3 G the authors report electron micrography of CAG-TFAM hearts, showing less densely packed cristae. Apart from that this data seems to contradict the substantial lack of the effect of

increased TFAM level in the heart of CA-TFAM mice (the authors should comment this point), there are no data about the other tissues and in particular for the skeletal muscle. Since one of the points that the authors stress concerns the different behaviour of the three tissues examined against the increased level of TFAM, it should be useful to report ultrastructural data also for them. In addition the analysis of the nucleoid clustering is reported only in the heart, while no data are presented for the skeletal muscle. Also in this case the authors should provide evidence concerning the situation in this organ.

6. The authors hypothesize that LONP1 degrades free TFAM in liver and that this is the reason for the lower level of the protein in the liver of CAG-TFAM mice compared to heart and skeletal muscle. Since TFAM degradation takes place after the protein is phosphorylated, they test the phosphorylation state of the skeletal muscle protein and did not find any evidence of increased phosphorylation. While this data might explain the high level of TFAM in this tissue (on the other hand also the level of LONP1 is low in skeletal muscle), it lacks the most important data, i.e the phosphorylation state of the protein in the liver. The authors should provide this data that would support their hypothesis.

7. The Fig 4D reporting the organelle assay for the mtDNA replication in CAG-TFAM mice is of low quality; bands are very faint and I think that it is not possible conclude from the figure that the CAG-TFAM exhibit a drastic decrease of mtDNA replication. Although the 7S DNA band is stronger for the control (longer exposure) the CAG-TFAM mice show a more intense signal along the gel. The authors should repeat this experiment or they may consider to omit it and present a more careful study elsewhere.

MINOR POINTS

1. FIG 1B: The authors should indicate (in the legend or in the figure) that the data concern the BAC TG137 line

2. In most of the histograms the authors use a similar colour code (blue-green) for the different types of reported data. This makes a bit confusing for the reader to distinguish between the different samples. Perhaps the use of different colour codes (yellow, red..) may be useful

Reviewer #2 (Comments to the Authors (Required)):

High levels of TFAM repress mammalian mitochondrial DNA transcription in Vivo

The abstract accurately describes the aims and outcomes of the investigation and the introduction is very well written laying out the key background and context of the investigation.

The experiments are performed well and the data is neatly presented. The observations are an important contribution to the field.

The point "moderate overexpression of TFAM of 1.63-fold in heart, 1.50-fold in liver and 1.49-fold in skeletal muscle in the BAC TG 137 line (Figure 1B). The other founder lines displayed a very similar increase in TFAM levels (Figure EV1A, B)." does not look quite accurate. Fig1B shows the biggest increase in TFAM expression in heart whereas the 2 examples in EV1A show the highest levels in liver, and the change in sk musc looks negligible TG188 when taking into account the loading as represented by b-actin.

Page 8 - "small number of COX-negative cardiomyocytes in the heart (Figure 2F)." The authors are

very conservative here as the image shown has possibly as much as 20% COX deficient cells.

Page 9 - "mitochondrial targeting sequence showed no increase in non-imported precursor proteins in liver and skeletal muscle tissue extracts, even after prolonged exposure of blots (Figure EV3D)." It is true that there is no increase in these markers in CAG samples, however there does appear to be a substantial decrease in steady state amounts of NDUFA9 and ATP5A proteins in the skeletal muscle. Accounting for the tubulin loading control there may be a decrease in NDUFA9 in the liver sample. It is not clear that the trend seen for the ATP5A marker is the same in the data from the 3 CAG mice (Figs 3A, and EV3D). The conclusion that the decrease in mitochondrial function occurs without affecting mt-protein import does not appear to be fully substantiated by the data.

Page 9 - It is not quite clear what is meant by 'comparable' in this sentence "Although total TFAM protein levels were comparable in both heart and skeletal muscle in CAG-TFAM mice (Figure 2B),...". In fig 2B the intensity (blackness) of the TFAM signals are comparable between the 3 tissue samples but not similar if one regards the levels relative to the control as given in the ratios beneath the images. Similarly, the actin levels in each tissue are similar between control and CAG but not between tissues and so if the TFAM level is compared to the actin then these are not similar.

Fig 3A does not appear to have a loading control associated with it. SDHB does not look to be affected in sk Musc but if the argument is that mitochondrially imported proteins may be affected then a cytosolic control of ponceau staining would improve the validity of the conclusions (p9).

The steady state levels of mtDNA are investigated for heart and sk musc (fig 3C) but not liver. The mtDNA replication in organello is then determined for liver, preventing the comparison between steady state and de novo in the same tissue.

Imaging of the nucleoids in all 3 tissues rather than only cardiomyocytes (cardiac sections), to show the apparent differences in compaction and aggregation would have been helpful to support the suggestion in the discussion that "induction of both LONP1 and POLRMT expression in liver which may keep a larger fraction of nucleoids in an open, active state, allowing maintained gene expression."

The discussion would improve with some finessing and editing - The initial part of the discussion describing the compaction of the nuclear DNA is rather more extensive than necessary to make the point about TFAM, mtDNA and transcriptional gene expression. It doesn't explain how this conflicts with the increase in mtDNA levels in the high TFAM CAG heart tissue, despite mentioning that in vitro there is a repressive effect in reported in vitro experiments. Similarly, the information given on the yeast orthologue does not currently link well within the discussion.

Minor points

In the introduction - possibly remove 'of all' in the sentence "may affect all (homoplasmy) or only a subset of all (heteroplasmy) mtDNA molecules..."

Page 8 - typo "CAG-TFAM allele preceeded by loxP-flanked"

Reviewer #3 (Comments to the Authors (Required)):

The manuscript by Bonekamp and colleagues describes the use of transgenic mice to study the function of TFAM. TFAM is a critical transcription factor for mtDNA, but it also works binds to mtDNA to form a nucleo-protein complex. The authors used two sets of transgenic, expressing moderate or high levels of TFAM. Surprisingly, the high expressors showed a severe transcription impairment. The data suggests that the levels of TFAM promote an equilibrium between active and inactive nucleoids. They also found that LONP1 and POLRTMT can compensate for TFAM overexpression and promote the more active sate of the nucleoid. The take home message is that the levels of TFAM are critical for optimum, and likely dynamic mtDNA expression. They also found interesting differences regarding TFAM overexpression in different tissues.

The data is of high quality and I have no major concerns with the manuscript. One could argue that the new knowledge is somehow incremental. However, the fact that these were done in vivo and analyzed in different tissues, I believe the study provides important new information of the role of TFAM in mitochondrial gene expression.

One piece of data that I found very puzzling is the difference observed between heart and skeletal muscle, where the latter was severely affected by overexpression of TFAM whereas the former was not. Is there an explanation for that?

Point-by-Point Response:**Reviewer #1:**

The paper of Bonekamp et al deals with a very interesting topic: the effect of the TFAM level on the mitochondrial gene expression in vivo. The authors, use transgenic mice overexpressing TFAM and analyse the impact of the increased TFAM level on mitochondrial gene expression in different organs. They found that a moderate overexpression of TFAM maintains a normal TFAM/mtDNA ratio in all tissue examines (heart, skeletal muscle liver) and has no substantial effect on mitochondrial function and on the whole animal metabolism. Then the authors overexpress TFAM at a higher level and find a strong repression of mitochondrial function in skeletal muscle, associated with a strong increase of the TFAM/mtDNA. In heart and in liver however the TFAM/mtDNA ratio is almost normal and OXPHOS capacity is maintained. The authors argue that in liver the higher tissue specific expression of LONP1 protease counteracts the TFAM increase, suggesting the existence of tissue specific regulatory factors that counterbalance the TFAM-induced repression of mtDNA transcription. The data reported in the paper are convincing and clearly reported and in general the paper is well written. However there are a number of points that need to be considered before the paper can be considered for the publication

We would like to thank the reviewer for his/her kind words and for the careful assessment of our manuscript.

MAJOR POINTS

1. A first basic point concerns the way in which the data are presented . Initially the authors, describe the BAC-TFAM mice and show the level of TFAM, mtDNA, mtRNA respiratory complexes and then report data on the metabolism of the whole animal. Then they describe the CAG-TFAM mice in this case the data are presented in a very different order; in particular data concerning mtDNA, mtRNA and respiratory complexes are shown almost at the end of the paper. I think that this may confuse the reader and therefore I believe that for more clarity the authors should present the data in a more ordered way.

We agree with these comments and have now extended the number of figures and present our data in a more consistent order to improve clarity.

In addition, and most importantly, the authors have carried out a complex experimental plan reporting a variety of the effects of TFAM level in three tissues (heart, skeletal muscle and liver), While I certainly appreciate this effort, that has produced a considerable amount of data, I found that in some cases the data are rather fragmentary; some results are presented only in one or two tissues and this makes difficult to assess properly the significance of the paper. Therefore the authors should complete the experiments in the points specified below (see point 5 and point 6)
2. The authors report the steady state level of mitochondrial transcripts (Fig 1E,F, EV1E,F and Fig 3D,E) by using the Northern blot method. I think that the authors should present (at least in some cases) also data coming from qRT-PCR analyses. In particular the data shown in Fig 1E (liver) and 3D (skeletal muscle) are of low quality: bands are very faint and it is difficult to make any quantitative assessment.

We have performed qRT-PCR analyses of steady-state mitochondrial transcript levels as suggested. We replaced the Northern blot panels in the main figures and now instead show the qRT-PCR results in Figure 1 E (heart and liver of the BAC-TFAM founder TG137; +/-BAC) and in Figures 3E and 5E (heart, liver and skeletal muscle of CAG-TFAM mice).

3. In Fig 1I the authors report the respiratory exchange rate (RER) in BAC-TFAM mice and conclude that they were similar to control rats. This is true for all samples tested except for the male /day and male/night at 52 weeks, in which the value of BAC-TFAM mice is higher than controls. The authors should explain this difference.

We re-analyzed our data sets and indeed observed a statistically significant increase of the RER in BAC-TFAM males at 52 weeks, which indicates a possible increased utilization of carbohydrates in those animals. The increase of RER in males at 52 weeks does not correlate with any significant change in feeding behavior or increased activity. It is possible that a moderate increase of mtDNA copy number will impact aging and age-related changes in metabolism and we plan to address this topic in future studies where we will age large cohorts of mice with a moderate increase of mtDNA copy number.

4. To rule out that the mitochondrial dysfunction in BAC-TFAM mice was caused by an overaccumulation of mitochondrial precursors, the authors perform a western blot of mitochondrial matrix protein of liver and skeletal muscle and conclude that there is no accumulation of non-imported precursors (Fig EV3D). The authors state that the bands shown in the figure refer to unprocessed precursors; I think that to justify this statement they should show some MW marker and the corresponding mature products. Moreover, in Fig EV3 the authors present data on the distribution of ACOT2 in the heart demonstrating a similar pattern of distribution in mitochondria and cytosol of control and CAG-TFAM animals. I wonder why the authors have performed different tests in the three tissues examined; in my opinion (unless there is a specific reason) it would have been preferable to test the same protein in the three organs.

We apologize for the confusion. For our whole tissue western blot analysis we reasoned that a potential overaccumulation of mitochondrial precursors in the cytosol would lead to the detection of a double band for ATP5A and NDUFA9, corresponding to the unprocessed and mature protein isoforms, respectively. We concluded that the bands we find represent the mature form, not the unprocessed form. We have now repeated our western blot analysis for NDUFA9 and extended our analysis to other mitochondrial proteins described to show clearly detectable unprocessed forms in addition to the mature proteins upon import block (PMID: 29650645, 31630969). We further extended the analysis to all three CAG-TFAM tissues investigated; please note that we detect a substantial decrease of protein levels in skeletal muscle, corresponding to the respiratory chain defect we observe by mass spectrometry (Figures S3E, Figure 4). We also repeated the subcellular fractionation on liver tissue and got a similar result (Figure S3D). The method employed (Abcam Mitochondrial Isolation Kit for Tissue) did not work on skeletal muscle, likely because this tissue is very compact and hard to disrupt. To further substantiate our conclusions, we re-analyzed the whole cell mass spectrometry data to detect potential peptides corresponding to the mitochondrial targeting sequence (Figure S3F) and to quantify any possible general decline in the import of matrix proteins belonging to the TCA cycle, lipid metabolism, acetyl CoA metabolism, iron-sulfur cluster (FeS) synthesis and heme synthesis pathways (Figure S3G). We did not detect any peptides corresponding to the mitochondrial targeting sequence. Furthermore, we observed no decrease of matrix protein levels but rather an increase of some matrix proteins in the most affected tissue, i.e. skeletal muscle. Based on these results, we conclude that mitochondrial protein import is not perturbed by high levels of TFAM protein expression.

5. In Fig EV3 G the authors report electron micrography of CAG-TFAM hearts, showing less densely packed cristae. Apart from that this data seems to contradict the substantial lack of the effect of increased TFAM level in the heart of CA-TFAM mice (the authors should comment this point), there are no data about the other tissues and in particular for the skeletal muscle. Since one of the points that the authors stress concerns the different behaviour of the three tissues examined against the increased level of TFAM, it should be useful to report ultrastructural data also for them. In addition the analysis of the nucleoid clustering is reported only in the heart, while no data are presented for

the skeletal muscle. Also in this case the authors should provide evidence concerning the situation in this organ.

To address the issues raised by the reviewer, we have performed ultrastructural analysis of mitochondria in heart, liver and skeletal muscle in a larger cohort of control (n=6) and +/-CAG-TFAM (n=6) mice (Figures 3G, 5G). We observed some alteration in heart cristae morphology, whereas there were very severe aberrations of mitochondrial ultrastructure and tissue organization in skeletal muscle. Ultrastructural analysis of liver revealed an accumulation of lipid droplets.

Analysis of nucleoid morphology by STED of skeletal muscle and liver did not work in our hands. The tissue preservation was good enough for confocal microscopy, but lacked the quality need for STED imaging. As we cannot present STED results from all tissues, we have moved the STED analysis of heart to the supplementary data and included a sentence on this limitations of the study in the discussion.

6. The authors hypothesize that LONP1 degrades free TFAM in liver and that this is the reason for the lower level of the protein in the liver of CAG-TFAM mice compared to heart and skeletal muscle. Since TFAM degradation takes place after the protein is phosphorylated, they test the phosphorylation state of the skeletal muscle protein and did not find any evidence of increased phosphorylation. While this data might explain the high level of TFAM in this tissue (on the other hand also the level of LONP1 is low in skeletal muscle), it lacks the most important data, i.e the phosphorylation state of the protein in the liver. The authors should provide this data that would support their hypothesis.

We analyzed the phosphorylation state of TFAM in liver and skeletal muscle by PhosTag-PAGE, but did not detect any phosphorylation by band shift (Fig. S5B, C). We have extensively looked for TFAM phosphorylations in unpublished work using other experimental systems and so far we have not been able to verify the model that TFAM is phosphorylated prior to degradation. A limitation of the current manuscript is that phosphorylated TFAM, if it exists, may be rapidly degraded by LONP1 in liver from CAG-TFAM mice and that it therefore may be hard to detect. However, phosphorylated TFAM, if present, should be easier to detect in skeletal muscle because of the large free pool of TFAM and the barely detectable levels of LONP1. Based on the results in this manuscript and unpublished results from ongoing studies in our laboratory we feel that the role for TFAM phosphorylation as a mark for degradation must be questioned and needs further investigation.

7. The Fig 4D reporting the organelle assay for the mtDNA replication in CAG-TFAM mice is of low quality; bands are very faint and I think that it is not possible conclude from the figure that the CAG-TFAM exhibit a drastic decrease of mtDNA replication. Although the 7S DNA band is stronger for the control (longer exposure) the CAG-TFAM mice show a more intense signal along the gel. The authors should repeat this experiment or they may consider to omit it and present a more careful study elsewhere.

We have decided to omit this data set.

MINOR POINTS

1.FIG 1B: The authors should indicate (in the legend or in the figure) that the data concern the BAC TG137 line.

We have indicated the use of the BAC TG137 line as +/-BAC in the text and figure legend.

2. In most of the histograms the authors use a similar colour code (blue-green) for the different types of reported data. This makes a bit confusing for the reader to distinguish between the different

samples. Perhaps the use of different colour codes (yellow, red..) may be useful

We have changed the color code in the new Figure S1 and S2 to a blue-green-yellow code to distinguish between the founders. We have added a caption above the histograms for molecular analysis (TFAM, mtDNA, mtRNA) to indicate more clearly what is shown.

Reviewer #2:

High levels of TFAM repress mammalian mitochondrial DNA transcription in Vivo

The abstract accurately describes the aims and outcomes of the investigation and the introduction is very well written laying out the key background and context of the investigation. The experiments are performed well and the data is neatly presented. The observations are an important contribution to the field.

We would like to thank the reviewer for considering our manuscript an important contribution to the field.

The point "moderate overexpression of TFAM of 1.63-fold in heart, 1.50-fold in liver and 1.49-fold in skeletal muscle in the BAC TG 137 line (Figure 1B). The other founder lines displayed a very similar increase in TFAM levels (Figure EV1A, B)." does not look quite accurate. Fig1B shows the biggest increase in TFAM expression in heart whereas the 2 examples in EV1A show the highest levels in liver, and the change in sk musc looks negligible TG188 when taking into account the loading as represented by b-actin.

We now describe the differences in more detail in the text.

Page 8 - "small number of COX-negative cardiomyocytes in the heart (Figure 2F)." The authors are very conservative here as the image shown has possibly as much as 20% COX deficient cells.

We re-assessed the extent of mitochondrial dysfunction in heart tissue using an alternative enzyme histochemical method (NBTx, PMID: 29660116). We observe variations in the number of COX-deficient cardiomyocytes between the individual CAG animals and within different areas of the tissue. Some areas indeed display a deficiency of ca. 20%. We have now more carefully referred to a "smaller number of COX-negative cells" in the text as opposed to the severe mitochondrial dysfunction observed in skeletal muscle.

Page 9 - "mitochondrial targeting sequence showed no increase in non-imported precursor proteins in liver and skeletal muscle tissue extracts, even after prolonged exposure of blots (Figure EV3D)." It is true that there is no increase in these markers in CAG samples, however there does appear to be a substantial decrease in steady state amounts of NDUFA9 and ATP5A proteins in the skeletal muscle. Accounting for the tubulin loading control there may be a decrease in NDUFA9 in the liver sample. It is not clear that the trend seen for the ATP5A marker is the same in the data from the 3 CAG mice (Figs 3A, and EV3D). The conclusion that the decrease in mitochondrial function occurs without affecting mt-protein import does not appear to be fully substantiated by the data.

See comments to Reviewer #1, point 4.

Page 9 - It is not quite clear what is meant by 'comparable' in this sentence "Although total TFAM protein levels were comparable in both heart and skeletal muscle in CAG-TFAM mice (Figure 2B),...". In fig 2B the intensity (blackness) of the TFAM signals are comparable between the 3 tissue samples

but not similar if one regards the levels relative to the control as given in the ratios beneath the images. Similarly, the actin levels in each tissue are similar between control and CAG but not between tissues and so if the TFAM level is compared to the actin then these are not similar.

We apologize for the confusion; we are referring to the relative increase of TFAM expression (increase caused by overexpression in comparison with wild-type TFAM expression) in the different tissues. We have not determined absolute TFAM levels. We have clarified this in the text.

Fig 3A does not appear to have a loading control associated with it. SDHB does not look to be affected in sk Musc but if the argument is that mitochondrially imported proteins may be affected then a cytosolic control of ponceau staining would improve the validity of the conclusions (p9).

We have repeated the western blot in question using a Ponceau S staining as loading control as suggested.

The steady state levels of mtDNA are investigated for heart and sk musc (fig 3C) but not liver. The mtDNA replication in organello is then determined for liver, preventing the comparison between steady state and de novo in the same tissue.

We have now included the steady-state mtDNA levels for liver as determined by qRT-PCR and Southern blot (Figures 5C and D).

Imaging of the nucleoids in all 3 tissues rather than only cardiomyocytes (cardiac sections), to show the apparent differences in compaction and aggregation would have been helpful to support the suggestion in the discussion that "induction of both LONP1 and POLRMT expression in liver which may keep a larger fraction of nucleoids in an open, active state, allowing maintained gene expression."

Please refer to the comments to Reviewer #1, point 5, second paragraph.

The discussion would improve with some finessing and editing - The initial part of the discussion describing the compaction of the nuclear DNA is rather more extensive than necessary to make the point about TFAM, mtDNA and transcriptional gene expression. It doesn't explain how this conflicts with the increase in mtDNA levels in the high TFAM CAG heart tissue, despite mentioning that in vitro there is a repressive effect in reported in vitro experiments. Similarly, the information given on the yeast orthologue does not currently link well within the discussion.

We have re-written the discussion.

Minor points In the introduction - possibly remove 'of all' in the sentence "may affect all (homoplasmy) or only a subset of all (heteroplasmy) mtDNA molecules..."

Page 8 - typo "CAG-TFAM allele preceeded by loxP-flanked"

We have corrected the sentences in question.

Reviewer #3:

The manuscript by Bonekamp and colleagues describes the use of transgenic mice to study the function of TFAM. TFAM is a critical transcription factor for mtDNA, but it also works binds to mtDNA to form a nucleo-protein complex. The authors used two sets of transgenic, expressing moderate or high levels of TFAM. Surprisingly, the high expressors showed a severe transcription impairment. The

data suggests that the levels of TFAM promote an equilibrium between active and inactive nucleoids. They also found that LONP1 and POLRTMT can compensate for TFAM overexpression and promote the more active state of the nucleoid. The take home message is that the levels of TFAM are critical for optimum, and likely dynamic mtDNA expression. They also found interesting differences regarding TFAM overexpression in different tissues. The data is of high quality and I have no major concerns with the manuscript. One could argue that the new knowledge is somehow incremental. However, the fact that these were done in vivo and analyzed in different tissues, I believe the study provides important new information of the role of TFAM in mitochondrial gene expression. One piece of data that I found very puzzling is the difference observed between heart and skeletal muscle, where the latter was severely affected by overexpression of TFAM whereas the former was not. Is there an explanation for that?

We would like to thank the reviewer for his/her kind words about our manuscript.

We conclude that the difference observed between heart and skeletal muscle results from differences in the TFAM-to-mtDNA ratio. In skeletal muscle, we observe a strong increase in TFAM protein levels compared to controls without a concomitant increase in mtDNA levels, resulting in a highly increased TFAM-to-mtDNA ratio (Figure 3B, C). In vitro studies indicated that this results in nucleoid compaction and abolished mtDNA transcription and replication. In contrast to skeletal muscle, we observe markedly increased mtDNA levels in heart tissue of CAG-TFAM mice after high levels of TFAM expression. This leads to a far more balanced TFAM-to-mtDNA ratio in heart, which allows for continued mtDNA expression. The postnatal development of cardiomyocytes involves substantial mtDNA replication during the first four weeks of postnatal life (PMID: 31170154) which may explain the observed increase in mtDNA levels in heart. However, we observe milder effects of TFAM overexpression in the heart and we cannot rule out that more detrimental effects in heart may occur with increasing age. Recent studies have revealed functional resilience mechanisms in response to insults to mtDNA expression in heart (PMID: 33760663) which might also compensate the CAG phenotype initially.

August 17, 2021

RE: Life Science Alliance Manuscript #LSA-2021-01034-TR

Prof. Nils-Göran Larsson
Karolinska Institutet
Department of Medical Biochemistry and Biophysics
Sölnavägen 9
Stockholm, Stockholm 17165
Sweden

Dear Dr. Larsson,

Thank you for submitting your revised manuscript entitled "High levels of TFAM repress mammalian mitochondrial DNA transcription in vivo". We would be happy to publish your paper in Life Science Alliance pending final revisions necessary to meet our formatting guidelines.

- please make sure the author order in your manuscript and our system match
- we encourage you to revise the figure legend for figure 1 such that the figure panels are introduced in an alphabetical order
- please add the Twitter handle of your host institute/organization as well as your own or one of the first author in our system

LSA now encourages authors to provide a 30-60 second video where the study is briefly explained. We will use these videos on social media to promote the published paper and the presenting author. Corresponding or first-authors are welcome to submit the video. Please submit only one video per manuscript. The video can be emailed to contact@life-science-alliance.org

A. FINAL FILES:

-- High-resolution figure, supplementary figure and video files uploaded as individual files: See our

detailed guidelines for preparing your production-ready images, <https://www.life-science-alliance.org/authors>

B. MANUSCRIPT ORGANIZATION AND FORMATTING:

Sincerely,

Reviewer #2 (Comments to the Authors (Required)):

The authors have responded to the major points made by the reviewers and improved the quality of the manuscript considerably by doing so. In particular, the data has been extended in line with the 3 reviewers' comments. In addition, the quality of the data presented is much improved over the initial submission. Overall the flow of the text has been improved and all the concerns of this reviewer have been met.

August 20, 2021

RE: Life Science Alliance Manuscript #LSA-2021-01034-TRR

Prof. Nils-Göran Larsson
Karolinska Institutet
Department of Medical Biochemistry and Biophysics
Solnavägen 9
Stockholm, Stockholm 17165
Sweden

Dear Dr. Larsson,

Thank you for submitting your Research Article entitled "High levels of TFAM repress mammalian mitochondrial DNA transcription in vivo". It is a pleasure to let you know that your manuscript is now accepted for publication in Life Science Alliance. Congratulations on this interesting work.

DISTRIBUTION OF MATERIALS:

Again, congratulations on a very nice paper. I hope you found the review process to be constructive and are pleased with how the manuscript was handled editorially. We look forward to future exciting submissions from your lab.

Sincerely,
